# LightGTS: A Lightweight General Time Series Forecasting Model

Yihang Wang [* 1]  Yuying Qiu [* 1]  Peng Chen [1]  Yang Shu [1]  Zhongwen Rao [2]  Lujia Pan [2]  Bin Yang [1]
Chenjuan Guo [1]

## Abstract

Existing works on general time series forecasting build foundation models with heavy model parameters through large-scale multi-source pre-training. These models achieve superior generalization ability across various datasets at the cost of significant computational burdens and limitations in resource-constrained scenarios. This paper introduces LightGTS, a lightweight general time series forecasting model designed from the perspective of consistent periodical modeling. To handle diverse scales and intrinsic periods in multi-source pre-training, we introduce Periodical Tokenization, which extracts consistent periodic patterns across different datasets with varying scales. To better utilize the periodicity in the decoding process, we further introduce Periodical Parallel Decoding, which leverages historical tokens to improve forecasting. Based on the two techniques above which fully leverage the inductive bias of periods inherent in time series, LightGTS uses a lightweight model to achieve outstanding performance on general time series forecasting. It achieves state-of-the-art forecasting performance on 9 real-world benchmarks in both zero-shot and full-shot settings with much better efficiency compared with existing time series foundation models.

## 1. Introduction

Time series forecasting is widely applied across various domains, including energy, meteorology, education, finance, and transportation (Wu et al., 2021; 2023; 2024; 2025c). Traditional time series forecasting approaches typically use task-specific statistical or deep learning models in an end-to-end manner (Wu et al., 2025a; Qiu et al., 2025b; Wu et al., 2025b). Recently, with a collection of large-scale time series

---

[*]Equal contribution  [1]East China Normal University, Shanghai, China  [2]Huawei Noah's Ark Lab, Shenzhen, China. Correspondence to: Chenjuan Guo <cjguo@dase.ecnu.edu.cn>.

*Proceedings of the 42$^{nd}$ International Conference on Machine Learning*, Vancouver, Canada. PMLR 267, 2025. Copyright 2025 by the author(s).

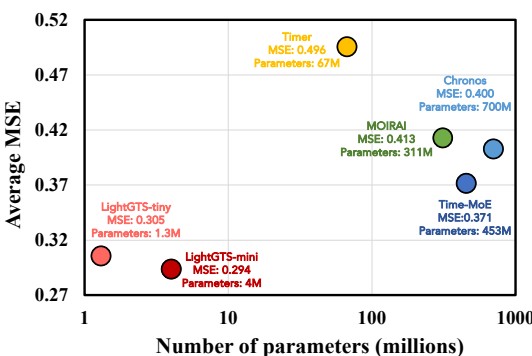

*Figure* 1. Comparison of model sizes and average zero-shot performance across seven benchmark datasets between LightGTS and the state-of-the-art TSFMs.

datasets, several Time Series Foundation Models (TSFM) have emerged (Liu et al., 2024; Woo et al., 2024a; Shi et al., 2024), demonstrating promising potential. However, the generalization capability of existing TSFMs largely depends on massive pre-training data and large model parameters as shown in Figure 1, resulting in high computational costs and low efficiency.

Although both language and time series are sequential data, unlike language which uses word tokens as basic elements, time series elements can vary substantially and exhibit distinctive characteristics. Specifically, *scale* refers to the sampling rate of time series, e.g., sampling every 15 mins or every hour, and *intrinsic period* reflects the time interval, e.g. daily, that a pattern repetitively appears in the real-world. Meanwhile, different scales affect the number of data points that appear in an intrinsic period, referred to as a *cycle length*, as shown in Figure 2(a). In multi-source time series pre-training, this scale-dependent variation necessitates models to learn consistent representations of real-world periodic patterns across different cycle lengths, a core demand for reliable forecasting as evidenced by the critical role of periodic modeling in (Lin et al., 2024b;a).

However, the existing TSFMs adopt fixed tokenization, where each token contains a fixed number of data points, making it struggle to handle diverse scales and intrinsic periods in multi-source pre-training. Specifically, fixed tokenization leads to varying information density of tokens

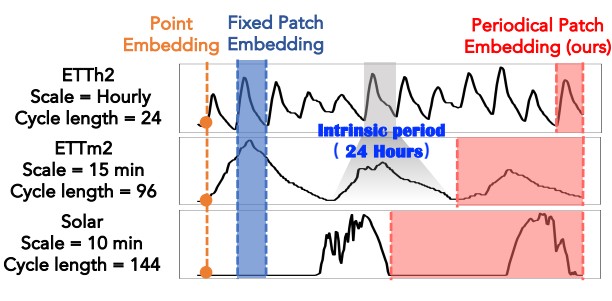
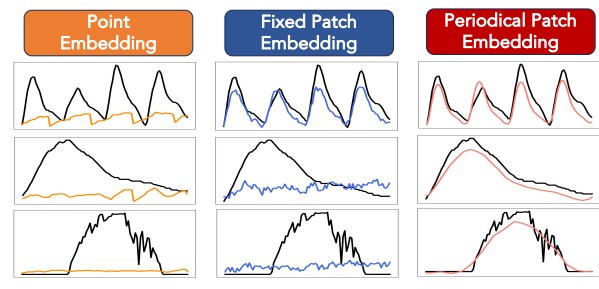

(a) The illustration of three tokenization methods

(b) The case study of training on ETTh1 (Hourly, Cycle length= 24) and testing on datasets with different scales and cycle lengths

*Figure* 2. **(a)** The illustration of three tokenization methods (Point embedding, Fixed patch embedding, Periodical patch embedding). For example, ETTh2 and ETTm2 share the same daily intrinsic period, but their cycle lengths differ due to differences in scale. **(b)** The case study of training on ETTh1 and testing on datasets with different scales and cycle lengths, all three tokenization methods recognized the intrinsic period on the same-scale datasets. However, only periodical tokenization successfully transferred to datasets with different scales.

across different scales, resulting in inconsistent feature representations. Additionally, it also disrupts the continuity and structural integrity of periodic patterns. To illustrate this point intuitively, we conducted a case study shown in Figure 2(b). After pre-training on a single-scale dataset, the model using the fixed tokenization recognizes the intrinsic period within the same scale, but significantly performs worse when transferred to datasets with different scales. This highlights how the fixed tokenization limits the model's ability to leverage time series inductive biases related to scale and intrinsic period, hindering the generalization ability and necessitating more parameters, which in turn increases computational costs and reduces efficiency.

In this paper, we aim to leverage the inherent inductive biases of scale-invariant intrinsic periods in time series data to design an efficient TSFM architecture. To achieve this, we propose **LightGTS**, which effectively utilizes such an inductive bias through adaptive periodical tokenization and periodical parallel decoding, compressing the model's parameter size while ensuring high performance.

To model intrinsic period consistently across different scales, we propose periodical tokenization, where the time series is adaptively divided into patches based on the cycle length, ensuring that each token captures a complete intrinsic period. Since the intrinsic period remains constant regardless of the scale, this approach ensures that the periodic patterns are consistently captured, and more importantly, the semantic information within each token remains aligned across different scales and cycle lengths. Furthermore, an embedding module with fixed projection cannot handle varying cycle lengths in multi-source time series pre-training, so we introduce a flex projection layer to address this issue. As shown in Figure 2(b), the periodical tokenization allows LightGTS to effectively generalize to datasets with varying scales and cycle lengths, improving the flexibility and adaptability of the model.

To better leverage the periodic patterns in the decoding process, we introduce a non-autoregressive decoding method, called periodical parallel decoding. Specifically, we initialize the decoder input with the last token generated by the encoder, consolidating all historical information. The key insight behind this design is that the last token not only maintains temporal continuity with future predictions but also leverages the structural consistency between historical and forecast horizons, enabling the effective extrapolation of periodic patterns and generating accurate predictions. In addition, the decoder outputs the predictions in parallel, which avoids cumulative errors and reduces computational cost.

The two techniques above enable the model to fully leverage the inductive bias inherent in time series data, significantly reducing the model's parameter size and lowering the computational resource requirements. As shown in Figure 1, LightGTS achieves state-of-the-art prediction performance with fewer than 5 million trainable parameters, making it 10 to 100 times smaller than its counterparts.

In summary, our contributions in this paper are as follows:

- We propose a novel periodical tokenization that adaptively splits patches based on the intrinsic period of the dataset, naturally extracting consistent periodic patterns across different scales.

- We further propose a simple and effective periodical parallel decoding which not only better leverages the periodicity of time series data but also avoids cumulative errors and reduces the computational cost.

- Based on the techniques above, we present Light-GTS, which fully leverages the inductive bias of time series data, making it achieve state-of-the-art predictive accuracy with less trainable parameters. Moreover, our code and pre-trained model check-points are available at https://github.com/decisionintelligence/LightGTS.

## 2. Related Work

### 2.1. Time Series Forecasting

Early time series forecasting (TSF) methods relied on statistical techniques like ARIMA (Box & Pierce, 1970) and VAR (Godahewa et al., 2021a). With machine learning advancements, methods such as GBoost (Chen & Guestrin, 2016) and LightGBM (Ke et al., 2017) emerged but required manual feature engineering. Leveraging the representation learning capabilities of deep neural networks (DNNs), models like Pathformer (Chen et al., 2024), and PatchTST (Nie et al., 2023) have surpassed traditional methods in forecasting accuracy. Recently, LLM-based methods like Time-LLM (Jin et al., 2023) and GPT4TS (Zhou et al., 2023) promise by leveraging LLMs' capabilities to capture complex time series patterns. Additionally, pre-training on multi-domain time series data has gained significant attention, with notable approaches such as MOIRAI (Woo et al., 2024b) and UniTS (Gao et al., 2024), demonstrating promising results.

### 2.2. Tokenization in Time Series Foundation Models

Tokenization strategies are essential for TSFMs to transform raw data into structured input. The two primary methods, Point Embedding and Fixed Patch Embedding, offer distinct advantages. Point Embedding based models (Ansari et al., 2024) encodes individual time steps as a distinct token, preserving fine-grained details for short-term dependencies but struggling with long-term patterns and noise. Fixed Patch Embedding segments the time series into fixed-length patches and embeds each patch as a single token. This tokenization technique improves computational efficiency, and captures aggregated patterns, making it well-suited for long-term dependencies and Transformer-based models such as Timer (Liu et al., 2024) and TimesFM (Das et al., 2023). However, neither the fixed tokenization above can deal with the varying scales and intrinsic periods in multi-source time series pre-training, limiting the generalization of the TSFMs.

### 2.3. Decoding in Time Series Foundation Models

Decoding strategies influence how TSFMs generate future values. Key approaches include: i) Autoregressive decoding (Shi et al., 2024; Ansari et al., 2024), where models predict sequentially, but suffer from slow inference and error accumulation. ii) MLP decoding, which predicts all future values simultaneously using flatten head, allowing parallel computation, but lacking flexibility for arbitrary forecasting horizons. iii) Masked autoencoders (Goswami et al., 2024; Gao et al., 2024), which train models to reconstruct missing values, improving temporal representation but relying on reconstruction loss and requiring large-scale data. Unlike these, our periodical parallel decoding combines flexibility and efficiency, while also making better use of the periodicity in time series data.

## 3. Methodology

**Problem Formulation** Given a multivariate time series $\mathbf{X}_t = \{\mathbf{x}^i_{t-L:t}\}^C_{i=1}$, where each $\mathbf{x}^i_{t-L:t} \in \mathbb{R}^L$ is a sequence of observations. $L$ denotes the look-back window and $C$ denotes the number of channels. The forecasting task is to predict the future values $\hat{\mathbf{Y}}_t = \{\hat{\mathbf{x}}^i_{t:t+F}\}^C_{i=1}$, where $F$ denotes the forecast horizon. $\mathbf{Y}_t = \{\mathbf{x}^i_{t:t+F}\}^C_{i=1}$ is the ground truth. The general time series forecasting model is pre-trained with multi-source datasets $\mathbf{D}_{\text{pre-train}} = \{(\mathbf{X}^j_t, \mathbf{Y}^j_t)\}^N_{j=1}$, where $N$ is the number of datasets. For the downstream task, the model is fine-tuned with a training dataset $\mathbf{D}_{\text{train}} = \{(\mathbf{X}^{\text{train}}_t, \mathbf{Y}^{\text{train}}_t)\}$, and is tested with $\mathbf{D}_{\text{test}} = \{(\mathbf{X}^{\text{test}}_t, \mathbf{Y}^{\text{test}}_t)\}$ to predict $\hat{\mathbf{Y}}^{\text{test}}_t$, where $\mathbf{D}_{\text{pre-train}}$, $\mathbf{D}_{\text{train}}$ and $\mathbf{D}_{\text{test}}$ are pairwise disjoint. Alternatively, the model could be directly tested using $\mathbf{D}_{\text{test}}$ without fine-tuning with $\mathbf{D}_{\text{train}}$ to predict $\hat{\mathbf{Y}}^{\text{test}}_t$.

### 3.1. Architecture

As shown in Figure 3, LightGTS enhances time series modeling through periodical tokenization, which comprises two coordinated components: (1) adaptive Periodical Patching to segment sequences with different scales into period patches, and (2) the Flex Projection Layer to embed period patches with various lengths into a shared semantic space. Integrated with the Transformer Encoder-Decoder, this approach first identifies intrinsic periodicities to generate period patches. The Flex Projection Layer then dynamically adjusts patch embedding weights, transforming diverse-length patches into dimensionally consistent tokens while retaining periodic semantics.

Once the patches are embedded into tokens, the tokens are then passed into the Transformer encoder to model the periodical patterns within the input time series. Notably, instead of relying on autoregressive methods, we employ a periodical parallel decoding approach(PPD) to construct the input for the Transformer decoder. This not only allows for more efficient and parallelizable processing of the sequence, but also better accommodates periodic modeling by automatically aligning periodic features between input and output sequences. Finally, the tokens generated by the Transformer decoder are transformed into prediction results.

3.1.1. PERIODICAL TOKENIZATION

**Periodical Patching:** For the simplicity of notations, we use a univariate time series for description, and the method can be easily extended to multivariate cases by considering each variable independently. Given a collection of time series from multi-source datasets where each $\mathbf{x} \in \mathbb{R}^L$ exhibits its own intrinsic period during pre-training, we first identify the cycle length $P$ (data points per intrinsic period) for each $\mathbf{x}$ via periods-finding.

$$P = \text{PeriodsFinding}(\mathbf{x}) \qquad (1)$$

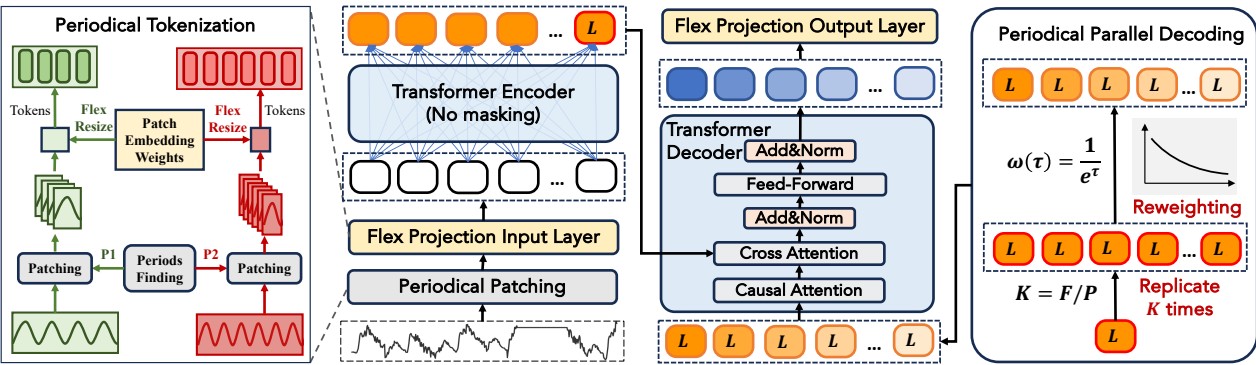

*Figure* 3. LightGTS architecture.

It is worth noting when prior knowledge of the input series is available, the cycle length can be inferred using information such as the sampling rate. In the absence of prior knowledge, the cycle length can also be deduced using methods such as Fast Fourier transform (FFT) (Wu et al., 2022).

After obtaining the cycle length $P$, the input series $\mathbf{x}$ is segmented into non-overlapping period patches $\mathbf{X}_p \in \mathbb{R}^{P \times N}$, where $N$ is the number of the patches, $N = \lfloor L/P \rfloor$. Each patch aligns precisely with the data's intrinsic periodic interval, ensuring that one patch encapsulates a full cycle. By doing so, time series of varying scales are interpreted within a unified period-aligned semantic space, eliminating interference from varying sampling rates and enabling coherent cross-source pattern learning during pre-training.

**Flex Projection Layer:** Each period patch preserves local periodic semantic information after segmenting the time series using the Period Patching mechanism. Then, the patch embedding projection transforms the period patches into tokens. However, time series from multi-source datasets exhibit diverse intrinsic periods and scales, leading to significant variation in patch sizes. Therefore, the fixed patch embedding projection struggles to process varying patch sizes. A straightforward approach is to resize the weights of patch embedding projection directly using methods like linear interpolation to accommodate different patch sizes. However, the simple linear interpolation may introduce biases into the resulting tokens, which may degrade model performance, as discussed below.

Consider processing the same dataset with two different scales, periodical patching on the dataset yields patches of two different sizes: $\mathbf{x} \in \mathbb{R}^P$ and $\mathbf{x}' \in \mathbb{R}^{P'}$. Based on the scale difference, we assume a linear interpolation relationship between these patches and have $\mathbf{x}' = \text{Interp}_P^{P'}(\mathbf{x})$. Suppose we already have the original patch embedding weights $\theta \in \mathbb{R}^{P \times D}$ tailored for projecting $\mathbf{x}$. Then we want to get the embedding projection weights $\theta'$ for $\mathbf{x}'$ that satisfies $\mathbf{x} \cdot \theta = \mathbf{x}' \cdot \theta'$. In a straightforward way, the original $\theta$ should be resized through interpolation to obtain $\theta' = \text{Interp}_P^{P'}(\theta) \in \mathbb{R}^{P' \times D}$ to

handle the input $\mathbf{x}'$ with a different patch size $P'$. However, this simple linear interpolation in embedding projection introduces substantial distortions in token representations and causes $\mathbf{x} \cdot \theta \neq \mathbf{x}' \cdot \theta'$. Such changes hinder the ability to adapt a pre-trained backbone, designed for a fixed patch size, to process inputs with varying patch sizes effectively.

We then discuss the proposed Flex Projection to address such inconsistency of token representations across different scales. Note that the linear interpolation can be formalized as a linear transformation:

$$\text{Interp}(\mathbf{x})_P^{P'} = \mathbf{x} \cdot \mathbf{A}, \tag{2}$$

where $\mathbf{A} \in \mathbb{R}^{P \times P'}$ represents the linear transformation matrix that resizes a vector $\mathbf{x}$ with length $P$ to length $P'$ by linear interpolation. The ideal objective of Flex Projection is to preserve token consistency across scales by deriving adjusted embedding weights $\theta'$ that satisfy $\mathbf{x} \cdot \theta = \mathbf{x}' \cdot \theta'$ when processing inputs at varying scales. Thus, we hope to find a new set $\theta'$ under the following optimization objective:

$$\theta' = \arg\min_{\theta'} \mathbb{E}_{x \sim \mathcal{X}} [||\mathbf{x} \cdot \theta - \mathbf{x}\mathbf{A} \cdot \theta'||_F^2], \tag{3}$$

where $\mathcal{X}$ is some distribution over the input patches, $|| \cdot ||_F$ is the Frobenius norm of a vector. In Section 3.2, we extend our analysis to account for shifts in patch distributions and theoretically demonstrate that the adjusted embedding weights must satisfy $\theta' = \delta^{-1}(\mathbf{A})^+ \theta$ to preserve token consistency. The term $\delta = \sqrt{\frac{P}{P'}}$ is the upper bound of the variation in the variances between $\mathbf{x}$ and $\mathbf{x} \cdot \mathbf{A}$, where $()^+$ denotes the Moore-Penrose pseudoinverse of the matrix. Thus, we define flex-resize as:

$$\text{Flex-resize}(\theta)_P^{P'} = \delta^{-1}(\mathbf{A})^+ \theta \tag{4}$$

The flex-resize ensures the equivalence of patch embeddings across different patch sizes while preserving the stability of statistical characteristics. Leveraging the flex-resize, we propose the Flex Projection Layer which adapts to varying cycle lengths without compromising the performance.

Specifically, we define two underlying parameter matrices for the input layer and output layer: $\theta_e \in \mathbb{R}^{P^* \times D}$ and $\theta_d \in \mathbb{R}^{P^* \times D}$, where $P^*$ represents a predefined reference patch size, and $D$ is the latent dimension. These learnable parameters are dynamically resized with the resize transformation $\delta^{-1}(\mathbf{A})^+ \in \mathbb{R}^{P^* \times P}$ during the forward pass to match the required patch size for each time series. Using the resized patch embedding, we then map the patches $\mathbf{X}_p$ into the Transformer latent space to obtain the tokens $\mathbf{X}_e \in \mathbb{R}^{D \times N}$:

$$\mathbf{X}_e = \mathbf{X}_p \cdot \text{Flex-resize}(\theta_e)_{P^*}^P, \tag{5}$$

### 3.1.2. ENCODING AND DECODING

Similarly to the original Transformer, our encoder and decoder modules are composed of Feed-Forward Networks and Multi-Head Attention modules. Notably, we leverage Rotary Positional Encoding (RoPE) (Su et al., 2021) in the attention modules to enhance the modeling of relative positional information between tokens.

**Encoding:** After obtaining the projected tokens $\mathbf{X}_e = \{\mathbf{x}_e^1, \mathbf{x}_e^2, ... \mathbf{x}_e^N\} \in \mathbb{R}^{D \times N}$, where the superscript $i$ in $\mathbf{x}_e^i$ denotes the i-th token, we incorporate RoPE into the attention mechanism to represent the relative positional information across different tokens. For brevity, we leave layers, attention head indices, and the scaling factor. Let $\mathbf{x}_e^i$ and $\mathbf{x}_e^j$ be the query and key vectors at positions $i$ and $j$, respectively. The transformed query and key vectors are $\mathbf{W}_Q \mathbf{x}_e^i$ and $\mathbf{W}_K \mathbf{x}_e^j$, where $\mathbf{W}_Q \in \mathbb{R}^{D \times D}$ and $\mathbf{W}_K \in \mathbb{R}^{D \times D}$ are learnable projections for queries and keys. The rotation matrix $\mathbf{R}_{i-j} \in \mathbb{R}^{N \times N}$ is then applied to adjust their relationship, and the similarity score is computed as:

$$\mathbf{S}_{ij} = (\mathbf{W}_Q \mathbf{x}_e^i)^T \mathbf{R}_{i-j} (\mathbf{W}_K \mathbf{x}_e^j), \tag{6}$$

The similarity scores $\mathbf{S}_{ij} \in \mathbb{R}$ are then normalized using the Softmax function and combined with the value vectors $\mathbf{W}_V \mathbf{x}_e^j$ to compute the attention output, where $\mathbf{W}_V \in \mathbb{R}^{D \times D}$.

$$\mathbf{Attn}_i = \sum_j \frac{\exp\{\mathbf{S}_{ij}\}}{\sum_k \exp\{\mathbf{S}_{ik}\}} (\mathbf{W}_V \mathbf{x}_e^j), \tag{7}$$

Finally, each token is processed through a Feed-Forward Network (FFN) to extract features, resulting in the latent representation $\mathbf{E} = \{\mathbf{e}_j\} \in \mathbb{R}^{D \times N}, j = 1, \ldots, N$.

**Periodical Parallel Decoding:** Periodical tokenization effectively extracts periodical information. To further leverage the periodical information in the decoding process, we propose periodical parallel decoding, a novel non-autoregressive decoding method. Given that the last token generated by the encoder not only retains the relevant intrinsic periodic characteristics but also consolidates historical information, we use it as the decoder's input to guide prediction process.

In contrast to autoregressive decoding, we replicate the last token of the latent representation, denoted as $\mathbf{e}_N \in$

$\mathbb{R}^{D \times 1}$, resulting in $\mathbf{H} = \{\mathbf{h}_j\} \in \mathbb{R}^{D \times K}, j = 1, ..., K$, where $K = \lceil F/P \rceil$ is the number of tokens corresponding to the prediction length. Additionally, the influence of the last token on the predicted sequence decreases as the prediction length grows. To account for this, we apply a reweighting function $\omega(\tau)$ to the tokens in $\mathbf{H}$, where $\tau$ denotes the position index of each token. Thus, the set of tokens $\mathbf{H}$ is weighted as $\omega(j) \mathbf{h}_j$. Ultimately, all tokens are simultaneously input into the Decoder:

$$\mathbf{Z} = \text{Decoder}\left(\{\omega(j) \mathbf{h}_j\}, \mathbf{E}\right), \quad \omega(\tau) = \frac{1}{e^\tau}, \tag{8}$$

$$\hat{\mathbf{Y}} = \text{Flex-resize}(\theta_d)_P^{P^*} \cdot \mathbf{Z}, \tag{9}$$

where $\mathbf{Z} \in \mathbb{R}^{D \times K}$ is the output of the Decoder. Finally, predictions are obtained through the Flex Projection Layer.

### 3.1.3. LOSS FUNCTIONS

In alignment with current mainstream practices in the field, we adopt the classic Mean Squared Error (MSE) as the loss function for LightGTS. This function measures the discrepancy between the predicted values $\hat{\mathbf{Y}}$ and the actual ground truth $\mathbf{Y}$. It is formulated as:

$$\mathcal{L}_{\text{MSE}} = ||\mathbf{Y} - \hat{\mathbf{Y}}||_F^2. \tag{10}$$

### 3.2. Theoretical Analysis

In Section 3.1.1, we have described that linear interpolation may change the effectiveness of patching embedding:

$$\text{Interp}(\mathbf{x})_P^{P'} = \mathbf{x} \cdot \mathbf{A}, \tag{11}$$

where $\mathbf{A} \in \mathbb{R}^{P \times P'}$ represents the linear mapping matrix. This operation resizes a vector with length $P$ to length $P'$ by linear interpolation. To pursue the equivalence of patching embedding, we hope to find a new set of $\theta'$ under the optimization problem defined by Theorem 3.1.

**Theorem 3.1.** *Patching embedding can be treated as a linear projection on patch $\mathbf{x}$ with parameters $\theta$. To preserve token consistency across scales, $\theta'$ is solved for keeping the minimum Euclidean distance between projected vectors:*

$$\theta' = \arg\min_{\theta'} \mathbb{E}_{\mathbf{x} \sim \mathcal{X}}[||\mathbf{x} \cdot \theta - \mathbf{x}\mathbf{A} \cdot \theta'||_F^2], \tag{12}$$

where $\mathcal{X}$ is some distribution over the patches, $|| \cdot ||_F$ is the Frobenius norm of a vector. Furthermore, to keep the distributional consistency, we refine the optimization problem as Proposition 3.2.

**Proposition 3.2.** *Time series pre-trained models utilize normalization during pretraining, thus they learn fixed distribution over patches which is susceptible to slight effects from interpolation. Therefore, additional normalization is*

*needed to align the distributions of projected patches with the pre-trained backbones:*

$$\theta' = \arg\min_{\theta'}\mathbb{E}_{\mathbf{x}\sim\mathcal{X}}[||\mathbf{x}\cdot\theta - norm(\mathbf{x}\mathbf{A})\cdot\theta'||_F^2], \quad (13)$$

*Since Revin normalization ($\mathcal{X} = \mathcal{N}(0, I)$) and linear interpolation does not change the mean values, the norm operation can be further expressed as multiplying by a constant $\delta$ to eliminate the variation in variances:*

$$\theta' = \arg\min_{\theta'}\mathbb{E}_{\mathbf{x}\sim\mathcal{X}}[||\mathbf{x}\cdot\theta - \delta\mathbf{x}\mathbf{A}\cdot\theta'||_F^2] \quad (14)$$

$$= \arg\min_{\theta'}\mathbb{E}_{\mathbf{x}\sim\mathcal{X}}[||\mathbf{x}\cdot(\theta - \delta\mathbf{A}\theta')||_F^2] \quad (15)$$

$$= \arg\min_{\theta'}\mathbb{E}_{\mathbf{x}\sim\mathcal{X}}[(\theta - \delta\mathbf{A}\theta')^T\mathbf{x}^T\mathbf{x}(\theta - \delta\mathbf{A}\theta')] \quad (16)$$

$$= \arg\min_{\theta'}(\theta - \delta\mathbf{A}\theta')^T\mathbb{E}_{\mathbf{x}\sim\mathcal{X}}[\mathbf{x}\mathbf{x}^T](\theta - \delta\mathbf{A}\theta') \quad (17)$$

$$= \arg\min_{\theta'}||\theta - \delta\mathbf{A}\theta'||_{\Sigma}^2 \quad (18)$$

In Proposition 3.2, $E_{\mathbf{x}\sim\mathcal{X}}[\mathbf{x}\mathbf{x}^T]$ denotes the uncentered covariance matrix, $||v||_{\Sigma}^2 = v^T\Sigma v$ is the quadric form of $v$. Since we only consider the case $\mathcal{X} = \mathcal{N}(0, I)$, $E_{\mathbf{x}\sim\mathcal{X}}[\mathbf{x}\mathbf{x}^T]$ is also the covariance matrix. And the optimization objective can be formulated as a least squares solution to a linear system of equations:

$$\theta' = \arg\min_{\theta'}||\theta - \delta\mathbf{A}\theta'||_F^2. \quad (19)$$

Singular value decomposition (SVD) technique is widely applied to obtain approximate or exact solutions to the aforementioned optimization problem, with the general solution being related to the Moore-Penrose pseudoinverse:

$$\theta' = \delta^{-1}(\mathbf{A})^+\theta, \quad (20)$$

where the $\delta = \sqrt{\frac{P}{P'}}$ is the upper bound of the variation in the variances between $\mathbf{x}$ and $\mathbf{x}\cdot\mathbf{A}$.

In summary, the process can be treated as conducting a resize transformation $\delta^{-1}(\mathbf{A})^+$ on the weights $\theta$ of projection layers. It works in a no-learning way by solving the optimization problem above, which saves computational overhead.

# 4. Experiments

## 4.1. Experimental Setup

**Pre-training datasets.** To pre-train a general time series forecasting model effectively, we collect a substantial number of publicly available datasets spanning diverse domains, including energy, nature, health, transportation, web, and economics. Detailed information about these datasets is provided in Appendix A.1.

**Evaluation datasets.** To ensure comprehensive and fair comparisons across different models, we conduct experiments

on nine widely recognized forecasting benchmarks as target datasets, all strictly exclusive from the pre-training datasets. These benchmarks include Weather, Traffic, Electricity, Solar, Exchange, and the four subsets of ETT, spanning multiple domains to validate model generalizability.

**Baselines.** We select five foundation models for comparison in zero-shot setting, including Timer (Liu et al., 2024), MOIRAI (Woo et al., 2024b), Chronos (Ansari et al., 2024), TimesFM (Das et al., 2023), Time-MoE (Goswami et al., 2024). We also select state-of-the-art deep time series models as baselines in full-shot setting, including PDF (Dai et al., 2024), iTransformer (Liu et al., 2023), Pathformer (Chen et al., 2024), FITS (Xu et al., 2023), TimeMixer (Wang et al., 2024a), and PatchTST (Nie et al., 2022).

**Setup.** Following prior studies, we use Mean Squared Error (MSE) and Mean Absolute Error (MAE) as evaluation metrics. All methods predict future values for lengths $F = \{96, 192, 336, 720\}$. We have pre-trained two variants of LightGTS: LightGTS-tiny with 1 million parameters and LightGTS-mini with 4 million parameters. Additional implementation details are provided in Appendix A.

## 4.2. Zero-shot Forecasting

To ensure a fair comparison, we conduct zero-shot predictions for each foundational model on downstream datasets not included in their pre-training data. As shown in Table 1, LightGTS-mini consistently achieves the state-of-the-art performance, delivering an average MSE reduction of over 30% compared to the most competitive baselines. Remarkably, even with fewer parameters, LightGTS-tiny still outperforms across the majority of datasets, achieving an average MSE reduction of 27%. Furthermore, LightGTS demonstrates superior performance compared to baselines with hundreds of millions of parameters. Specifically, it achieves average MSE reductions of 27% and 28% compared to Chronos and MOIRAI, respectively, and a relative improvement of 17% over Time-MoE. ***This highlights LightGTS's ability to effectively exploit the inherent inductive biases of time series data, enabling exceptional performance even with significantly fewer parameters.***

## 4.3. Full-shot Forecasting

As shown in Table 2, we present the results of the LightGTS in full-shot and zero-shot settings, and compare with other baselines in full-shot setting. Key observations are summarized as follows. First, as a general forecasting model, LightGTS achieves superior performance compared to the six state-of-the-art baselines with full-data training, achieving an average MSE reduction of 7%. Second, we observe that LightGTS in zero-shot setting significantly outperforms the baselines in full-shot setting across five datasets. This observation validates the strong transferability of LightGTS pre-trained on large multi-source data.

*Table* 1. Full results of zero-shot forecasting experiments. The average results of all predicted lengths are listed here. Lower MSE or MAE values indicate better predictions. A dash ('-') denotes datasets included in the model's pretraining and therefore excluded from testing. **Red**: the best, Blue: the 2nd best.

| Models | LightGTS-tiny | | LightGTS-mini | | Timer (2024) | | MOIRAI (2024) | | Chronos (2024) | | TimesFM (2024) | | Time-MoE (2025) | |
|---|---|---|---|---|---|---|---|---|---|---|---|---|---|---|
| Metric | MSE | MAE | MSE | MAE | MSE | MAE | MSE | MAE | MSE | MAE | MSE | MAE | MSE | MAE |
| ETTm1 | 0.345 | 0.378 | **0.327** | **0.37** | 0.768 | 0.568 | 0.39 | 0.389 | 0.551 | 0.453 | 0.435 | 0.418 | 0.376 | 0.406 |
| ETTm2 | 0.249 | 0.318 | **0.247** | **0.316** | 0.315 | 0.356 | 0.276 | 0.32 | 0.293 | 0.331 | 0.347 | 0.36 | 0.315 | 0.365 |
| ETTh1 | 0.401 | 0.424 | **0.388** | **0.419** | 0.562 | 0.483 | 0.51 | 0.469 | 0.533 | 0.452 | 0.479 | 0.442 | 0.394 | 0.420 |
| ETTh2 | 0.362 | 0.397 | **0.348** | 0.395 | 0.370 | 0.400 | 0.354 | **0.377** | 0.392 | 0.397 | 0.400 | 0.403 | 0.403 | 0.415 |
| Traffic | 0.610 | 0.399 | **0.561** | **0.381** | 0.613 | 0.407 | - | - | 0.615 | 0.421 | - | - | - | - |
| Weather | 0.219 | 0.266 | **0.208** | **0.256** | 0.292 | 0.313 | 0.26 | 0.275 | 0.288 | 0.309 | - | - | 0.270 | 0.300 |
| Exchange | **0.345** | **0.395** | 0.347 | 0.396 | 0.392 | 0.425 | 0.385 | 0.417 | 0.370 | 0.412 | 0.390 | 0.417 | 0.432 | 0.454 |
| Solar | 0.219 | 0.305 | **0.191** | **0.271** | 0.771 | 0.604 | 0.714 | 0.704 | 0.393 | 0.319 | 0.500 | 0.397 | 0.411 | 0.428 |
| Electricity | 0.233 | 0.319 | 0.213 | 0.308 | 0.297 | 0.375 | **0.188** | **0.273** | - | - | - | - | - | - |

*Table* 2. The results of LightGTS-mini in zero-shot and full-shot setting and other baselines in full-shot setting. The average results of all predicted lengths are listed here. Lower MSE or MAE values indicate better predictions. **Red**: the best, Blue: the 2nd best.

| Models | LightGTS-mini (zero-shot) | | LightGTS-mini (full-shot) | | PDF (2024) | | iTransformer (2024) | | Pathformer (2024) | | FITS (2024) | | TimeMxier (2024) | | PatchTST (2023) | |
|---|---|---|---|---|---|---|---|---|---|---|---|---|---|---|---|---|
| Metric | MSE | MAE | MSE | MAE | MSE | MAE | MSE | MAE | MSE | MAE | MSE | MAE | MSE | MAE | MSE | MAE |
| ETTm1 | 0.327 | 0.370 | **0.321** | **0.361** | 0.342 | 0.376 | 0.347 | 0.378 | 0.357 | 0.375 | 0.357 | 0.377 | 0.356 | 0.380 | 0.349 | 0.381 |
| ETTm2 | 0.247 | 0.316 | **0.239** | **0.303** | 0.250 | 0.313 | 0.258 | 0.318 | 0.253 | 0.309 | 0.254 | 0.313 | 0.257 | 0.318 | 0.256 | 0.314 |
| ETTh1 | 0.388 | 0.419 | **0.388** | **0.413** | 0.407 | 0.426 | 0.440 | 0.445 | 0.417 | 0.426 | 0.408 | 0.427 | 0.427 | 0.441 | 0.419 | 0.436 |
| ETTh2 | 0.348 | 0.395 | **0.335** | **0.377** | 0.347 | 0.391 | 0.359 | 0.396 | 0.360 | 0.395 | 0.335 | 0.386 | 0.347 | 0.394 | 0.351 | 0.395 |
| Traffic | 0.561 | 0.381 | **0.393** | 0.259 | 0.395 | 0.270 | 0.397 | 0.281 | 0.416 | 0.264 | 0.429 | 0.302 | 0.410 | 0.279 | 0.397 | 0.275 |
| Weather | 0.208 | 0.256 | **0.207** | **0.244** | 0.227 | 0.263 | 0.232 | 0.270 | 0.225 | 0.258 | 0.244 | 0.281 | 0.225 | 0.263 | 0.224 | 0.261 |
| Exchange | 0.347 | 0.396 | 0.322 | 0.383 | 0.350 | 0.397 | **0.321** | 0.384 | 0.384 | 0.414 | 0.349 | 0.396 | 0.385 | 0.418 | 0.322 | 0.385 |
| Solar | 0.191 | 0.271 | **0.179** | **0.220** | 0.200 | 0.263 | 0.202 | 0.260 | 0.204 | 0.228 | 0.232 | 0.268 | 0.203 | 0.261 | 0.200 | 0.284 |
| Electricity | 0.213 | 0.308 | **0.156** | **0.248** | 0.160 | 0.253 | 0.163 | 0.258 | 0.168 | 0.261 | 0.169 | 0.265 | 0.185 | 0.284 | 0.171 | 0.270 |

## 4.4. Efficiency Advantages of LightGTS

In addition to its excellent prediction performance, another notable advantage of LightGTS is its lightweight nature. Previously, Figure 1 visualized the parameter-performance comparison of LightGTS with other TSFMs. Table 3 presents a more comprehensive efficiency comparison between Light-GTS and other TSFMs in terms of both static and runtime metrics. It is evident that LightGTS significantly outperforms other models in terms of static metrics such as the number of parameters and Multiply–Accumulate Operations (MACs), being over ten times smaller than the next best model. This characteristic allows LightGTS to be deployed on devices with limited computational resources. Furthermore, in terms of runtime metrics such as Max Memory and Inference Time, LightGTS significantly outperforms other TSFMs, rivaling existing small models (i.e., PatchTST).

*Table* 3. Static and runtime performance metrics of LightGTS and other baselines on the ETTm1 dataset, evaluated with a forecast horizon of 720 and a batch size of 1.

| Models | Parameters | MACs | Max Mem.(MB) | Inference Time (s) |
|---|---|---|---|---|
| Time-MoE | 453 M | 5252.9 G | 14131 | 2.13 |
| TimesFM | 200 M | 624.9 G | 1395 | 0.16 |
| Chronos | 700 M | 92327.9 G | 10269 | 34.33 |
| MOIRAI | 300 M | 97.36 G | 2009 | 0.1 |
| Timer | 67.4 M | 52.6 G | 1435 | 0.08 |
| PatchTST | 6.3 M | 225 M | **672** | **0.01** |
| LightGTS | **4 M** | **213 M** | 713 | **0.01** |

## 4.5. Ablation Studies

**Effectiveness of Periodical Tokenization** To validate the effectiveness of periodical tokenization, we conducted experiments comparing fixed patching and periodical patching on models with different decoding methods. The results, shown in Table 4, demonstrate that periodical patching consistently outperforms fixed patching across all datasets. Furthermore, periodical patching provides significant improvements regardless of the decoding method used, highlighting its strong generalization capability.

*Table* 4. Ablations on key components of LightGTS in zero-shot setting, including periodical tokenization, and periodical parallel decoding. The average results of all predicted lengths are listed.

| Dataset | | ETT-avg | | Weather | | Electricity | | Traffic | |
|---|---|---|---|---|---|---|---|---|---|
| Decoding | Patching | MSE | MAE | MSE | MAE | MSE | MAE | MSE | MAE |
| AR | Fixed | 0.442 | 0.430 | 0.265 | 0.293 | 0.231 | 0.326 | 0.630 | 0.411 |
| AR | Periodical | 0.341 | 0.384 | 0.226 | 0.270 | 0.229 | 0.319 | 0.634 | 0.410 |
| MAE | Fixed | 0.537 | 0.489 | 0.339 | 0.349 | 0.372 | 0.428 | 0.803 | 0.534 |
| MAE | Periodical | 0.388 | 0.417 | 0.260 | 0.301 | 0.322 | 0.392 | 0.746 | 0.484 |
| PPD | Fixed | 0.436 | 0.427 | 0.262 | 0.288 | 0.226 | 0.315 | 0.621 | 0.403 |
| PPD | Periodical | **0.328** | **0.375** | **0.208** | **0.256** | **0.213** | **0.308** | **0.561** | **0.381** |

**Effectiveness of Periodical Parallel Decoding** To further validate the effectiveness of our periodical parallel decoding (PPD) method, we conducted an ablation study by replacing it with mainstream autoregressive (AR) decoding and

masked autoencoder (MAE) decoding. The results, shown in Table 4, indicate that PPD consistently outperforms both AR and MAE decoding under both fixed patching and periodical patching strategies. Among the three methods, MAE achieved the poorest performance, likely due to a task gap between its upstream reconstruction objective and the downstream prediction task. Furthermore, we observed that PPD only slightly outperforms AR when using fixed patching, primarily because PPD avoids the accumulation of errors in long-step predictions. However, with periodical patching, the performance gain of PPD over AR becomes more pronounced, as PPD better exploits the inherent periodicity.

### 4.6. Model Analasis

**Impact of the Reference Patch Size**

LightGTG introduces a hyperparameter $P^*$ in the flex projection layer, namely the predefined reference patch size before resizing. To evaluate its impact on the performance of LightGTS, we conducted an ablation study using four different reference patch sizes. The experimental results presented in Table 5 demonstrate consistently stable performance across all evaluated patch sizes, indicating that the model exhibits strong insensitivity to the selection of $P^*$. Therefore, we set the default value to 48 in all experiments empirically.

*Table 5.* MSE results of LightGTS with varied hyperparameters of reference patch size $P^*$. The average results of all predicted lengths are listed here.

| Models | ETT-avg | | Weather | | Electricity | | Traffic | |
|---|---|---|---|---|---|---|---|---|
| Metric | MSE | MAE | MSE | MAE | MSE | MAE | MSE | MAE |
| $P^* = 24$ | 0.335 | 0.380 | 0.213 | 0.262 | 0.217 | 0.312 | 0.563 | 0.384 |
| $P^* = 48$ | 0.328 | 0.375 | 0.208 | 0.256 | 0.213 | 0.308 | 0.561 | 0.381 |
| $P^* = 96$ | 0.331 | 0.375 | 0.210 | 0.256 | 0.217 | 0.307 | 0.566 | 0.388 |
| $P^* = 192$ | 0.334 | 0.377 | 0.212 | 0.258 | 0.212 | 0.304 | 0.566 | 0.385 |

**Impact of the Replicated Token Choosing** Since the initialization of the decoder input is crucial for non-autoregressive decoding (Gu et al., 2017), we select the last encoder token as the input. To validate this choice, we experiment with replacing it using other tokens, such as the learnable embedding, the CLS token, and the mean of the encoder tokens. As shown in Table 6, the last encoder token yields the best performance, likely due to its better alignment with the periodicity and greater relevance to the prediction task.

*Table 6.* MSE results of LightGTS with different replicated tokens. The average results of all predicted lengths are listed here

| Dataset | ETT-avg | | Weather | | Electricity | | Traffic | |
|---|---|---|---|---|---|---|---|---|
| Metric | MSE | MAE | MSE | MAE | MSE | MAE | MSE | MAE |
| LightGTS-learn | 0.342 | 0.383 | 0.278 | 0.325 | 0.231 | 0.326 | 0.627 | 0.410 |
| LightGTS-cls | 0.343 | 0.385 | 0.341 | 0.371 | 0.234 | 0.329 | 0.634 | 0.415 |
| LightGTS-mean | 0.404 | 0.433 | 0.328 | 0.350 | 0.273 | 0.361 | 0.703 | 0.464 |
| LightGTS-last | **0.328** | **0.375** | **0.208** | **0.256** | **0.213** | **0.308** | **0.561** | **0.381** |

**Robustness across Different Resolutions** In our experiments, we evaluated Timer, Time-MoE, and LightGTS on

the same datasets (ETT1 and ETT2) with different sampling granularities while predicting the same time horizon (96 hours) in zero-shot setting. As shown in Figure 4, the results show that Timer and Time-MoE exhibit significant performance variation when faced with different sampling granularities, whereas LightGTS maintains stable performance. This further supports the idea that existing time series foundation models are unable to handle the variations in scale within time series datasets. In contrast, LightGTS, by understanding the time series from its intrinsic period, remains unaffected by changes in dataset scale, thus maintaining stable performance across different sampling granularities.

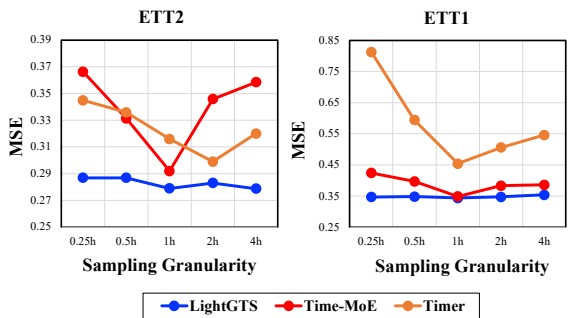

*Figure 4.* Comparisons of robustness across different sampling granularities between LightGTS, Timer, and Time-MoE in the zero-shot setting.

**Representation Learning of the Periodical Tokenization** As shown in Figure 5, we compare token representation similarity between Timer (fixed patch size=96) and LightGTS on solar with a daily intrinsic period under varying sampling granularities. At 10-minute sampling (cycle length=144), Timer's fixed patch fails to align with the full cycle, resulting in low inter-token similarity. Conversely, at 30-minute sampling (cycle length=48), patch size accidentally spans two full cycles, leading to high similarity. This inconsistency reveals the limitation of fixed tokenization in multi-scale temporal modeling. In contrast, LightGTS uses periodical tokenization, which produces the same number of tokens regardless of the sampling granularities, leading to consistent token representations across sampling granularities.

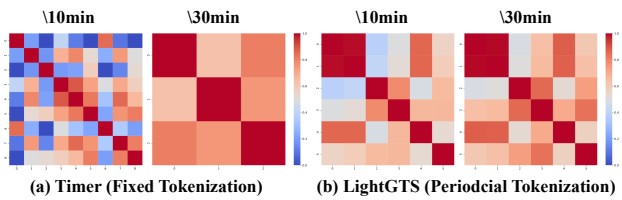

*Figure 5.* Similarity of token representation across various sampling granularities in Fixed (Timer) versus Periodical (LightGTS) Tokenization.

# 5. Discussion

## 5.1. LightGTS Compared to TTMs

TTMs incorporates novel hierarchical patch merging and resolution prefix tuning techniques to achieve better results. The LightGTS are indeed quite different from TTMs, including:

- **Flexibility:** TTMs have fixed input and output formats, which imposes limitations in downstream applications. In contrast, LightGTS supports flexible input and output configurations.

- **Adaptive Patching:** While TTMs employ adaptive patching through CV-inspired patch merging techniques to capture multi-scale features, they remain constrained by predefined patch sizes. LightGTS, however, leverages periodical patching that adaptively segments time series based on the intrinsic periods. This approach enables LightGTS to achieve unified modeling across datasets with varying scales.

## 5.2. LightGTS Compared to MOIRAI

While MOIRAI's predefined patch sizes based on sampling frequency offer some solutions for consistent modeling across different frequencies, they are still fixed and lack flexibility in certain scenarios. In contrast, Periodical Patching adaptively divides patches according to scale-invariant periodicity, enabling more flexible and unified modeling for datasets with varying frequencies.

# 6. Conclusion

In this work, we propose LightGTS, a lightweight general time series forecasting model leveraging the inductive bias of scale-invariant intrinsic periods in time series data. Light-GTS integrates adaptive periodical tokenization and periodical parallel decoding technique to enhance the generalization capability while maintain high efficiency. Our experiments show that LightGTS, using only 4 million parameters and achieving 10 to 100 times size reduction compared to counterparts, consistently attains superior performance in both zero-shot and full-shot settings.

# Impact Statement

This paper presents work whose goal is to advance the field of Machine Learning. There are many potential societal consequences of our work, none which we feel must be specifically highlighted here.

# Acknowledgements

This work was partially supported by National Natural Science Foundation of China (62372179, 62406112). Chenjuan Guo is the corresponding author of the work.

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

# A. Implementation Details

## A.1. Pre-training Datasets

We incorporate a diverse range of multi-source datasets for pre-training, which include portions from the Monash (Godahewa et al., 2021b), UEA (Bagnall et al., 2018), and UCR (Dau et al., 2019) time series datasets, as well as additional classic datasets (Zhang et al., 2017; Wang et al., 2024b; Liu et al., 2022; McCracken & Ng, 2016; Taieb et al., 2012). The complete list of pre-training datasets is shown in Table 7. It's important to note that there is no overlap between these pre-training datasets and the target datasets. Additionally, while the weather dataset in the pre-training set is univariate, the weather dataset in the target task is multivariate. The datasets used for pre-training are divided into six categories based on their domains: Energy, Nature, Health, Transport, and Web. The datasets exhibit a wide range of sampling frequencies, from millisecond intervals to monthly data, reflecting the complexity and variability of real-world applications. For all pre-training datasets, we split them into univariate sequences and train them in a channel-independent manner.

*Table* 7. List of pretraining datasets.

| Domain | Dataset | # Frequency | # Time Pionts | Source |
|---|---|---|---|---|
| Energy | Aus. Electricity Demand | Half Hourly | 1155264 | Monash (Godahewa et al., 2021b) |
| | Wind | 4 Seconds | 7397147 | Monash (Godahewa et al., 2021b) |
| | Wind Farms | Minutely | 172178060 | Monash (Godahewa et al., 2021b) |
| | Solar Power | 4 Seconds | 7397222 | Monash (Godahewa et al., 2021b) |
| | London Smart Meters | Half Hourly | 166527216 | Monash (Godahewa et al., 2021b) |
| Nature | Phoneme | - | 2160640 | UCR(Dau et al., 2019) |
| | EigenWorms | - | 27947136 | UEA (Bagnall et al., 2018) |
| | PRSA | Hourly | 4628448 | (Zhang et al., 2017) |
| | Temperature Rain | Daily | 23252200 | Monash (Godahewa et al., 2021b) |
| | StarLightCurves | - | 9457664 | UCR (Dau et al., 2019) |
| | Worms | 0.033 Seconds | 232200 | UCR (Dau et al., 2019) |
| | Saugeen River Flow | Daily | 23741 | Monash (Godahewa et al., 2021b) |
| | Sunspot | Daily | 73924 | Monash (Godahewa et al., 2021b) |
| | Weather | Daily | 43032000 | Monash (Godahewa et al., 2021b) |
| | KDD Cup 2018 | Daily | 2942364 | Monash(Godahewa et al., 2021b) |
| | US Births | Daily | 7305 | Monash (Godahewa et al., 2021b) |
| Health | MotorImagery | 0.001 Seconds | 72576000 | UEA (Bagnall et al., 2018) |
| | SelfRegulationSCP1 | 0.004 Seconds | 3015936 | UEA (Bagnall et al., 2018) |
| | SelfRegulationSCP2 | 0.004 Seconds | 3064320 | UEA (Bagnall et al., 2018) |
| | AtrialFibrillation | 0.008 Seconds | 38400 | UEA (Bagnall et al., 2018) |
| | PigArtPressure | - | 624000 | UCR (Dau et al., 2019) |
| | PIGCVP | - | 624000 | UCR (Dau et al., 2019) |
| | TDbrain | 0.002 Seconds | 79232703 | (Wang et al., 2024b) |
| Transport | Pems03 | 5 Minute | 9382464 | (Liu et al., 2022) |
| | Pems04 | 5 Minute | 5216544 | (Liu et al., 2022) |
| | Pems07 | 5 Minute | 24921792 | (Liu et al., 2022) |
| | Pems08 | 5 Minute | 3035520 | (Liu et al., 2022) |
| | Pems-bay | 5 Minute | 16937700 | (Liu et al., 2022) |
| | Pedestrian_Counts | Hourly | 3132346 | Monash (Godahewa et al., 2021b) |
| Web | Web Traffic | Daily | 116485589 | Monash (Godahewa et al., 2021b) |
| Economic | FRED_MD | Monthly | 77896 | (McCracken & Ng, 2016) |
| | Bitcoin | Daily | 75364 | Monash (Godahewa et al., 2021b) |
| | NN5 | Daily | 87801 | (Taieb et al., 2012) |

## A.2. Evaluation Datasets

We use the following 9 multivariate time-series datasets for downstream forecasting task: ETT datasets[1] contain 7 variates collected from two different electric transformers from July 2016 to July 2018. It consists of four subsets, of which ETTh1/ETTh2 are recorded hourly and ETTm1/ETTm2 are recorded every 15 minutes. Electricity[2] contains the electricity consumption of 321 customers from July 2016 to July 2019, recorded hourly. Solar[3] collects production from 137 PV plants in Alabama, recorded every 10 minutes. Traffic[4] contains road occupancy rates measured by 862 sensors on freeways in the San Francisco Bay Area from 2015 to 2016, recorded hourly. Weather[5] collects 21 meteorological indicators, such as temperature and barometric pressure, for Germany in 2020, recorded every 10 minutes. ExchangeRate[6] collects the daily exchange rates of 8 countries. We split each evaluation dataset into train-validation-test sets and detailed statistics of evaluation datasets are shown in Table 8.

*Table* 8. The statistics of evaluation datasets.

| Dataset | Domain | # Frequency | # Timestamps | # Split | # Dims | # Intrinsic Period | # Cycle Length |
|---------|--------|-------------|--------------|---------|--------|--------------------|----------------|
| ETTh1 | Energy | 1 hour | 14400 | 6:2:2 | 7 | Daily | 24 |
| ETTh2 | Energy | 1 hour | 14400 | 6:2:2 | 7 | Daily | 24 |
| ETTm1 | Energy | 15 mins | 57600 | 6:2:2 | 7 | Daily | 96 |
| ETTm2 | Energy | 15 mins | 57600 | 6:2:2 | 7 | Daily | 96 |
| Electricity | Energy | 10 mins | 26304 | 7:1:2 | 321 | Daily & Weekly | 24 & 168 |
| Solar | Energy | 10 mins | 52560 | 7:1:2 | 137 | Daily | 144 |
| Traffic | Traffic | 1 hour | 17544 | 7:1:2 | 862 | Daily & Weekly | 24 & 168 |
| Weather | Environment | 10 mins | 52696 | 7:1:2 | 21 | Daily | 144 |
| Exchange | Economic | 1 day | 7588 | 7:1:2 | 8 | - | - |

## A.3. Baselines

We select five foundation models for comparison in zero-shot setting, including Timer (Liu et al., 2024), MOIRAI (Woo et al., 2024b), Chronos (Ansari et al., 2024), TimesFM (Das et al., 2023), and Time-MoE (Goswami et al., 2024). In addition, we select the state-of-the-art models of deep time series models as our baselines in full-shot setting, including PDF , iTransformer (Liu et al., 2023), Pathformer (Chen et al., 2024), FITS (Xu et al., 2023), TimeMixer (Wang et al., 2024a), and PatchTST (Nie et al., 2022). The specific code base for these models is listed in Table 9:

*Table* 9. Code repositories for baselines.

| Model Types | Models | Code Repositories |
|-------------|--------|-------------------|
| Foundation model | Timer | https://github.com/thuml/Large-Time-Series-Model |
| | MOIRAI | https://github.com/redoules/moirai |
| | Chronos | https://github.com/amazon-science/chronos-forecasting |
| | TimesFM | https://github.com/google-research/timesfm |
| | Time-MoE | https://github.com/Time-MoE/Time-MoE |
| Small Model | PDF | https://github.com/Hank0626/PDF |
| | iTransformer | https://github.com/thuml/iTransformer |
| | Pathformer | https://github.com/decisionintelligence/pathformer |
| | FITS | https://github.com/VEWOXIC/FITS |
| | TimeMixer | https://github.com/kwuking/TimeMixer |
| | PatchTST | https://github.com/yuqinie98/PatchTST |

---

[1]https://github.com/zhouhaoyi/ETDataset

[2]https://archive.ics.uci.edu/ml/datasets/ElectricityLoadDiagrams20112014

[3]https://dl.acm.org/doi/abs/10.1145/3209978.3210006

[4]https://pems.dot.ca.gov/

[5]https://www.bgc-jena.mpg.de/wetter/

[6]https://dl.acm.org/doi/abs/10.1145/3209978.3210006

## A.4. Setting

**Pre-trainning** We implemented LightGTS using PyTorch (Paszke et al., 2019), and all experiments were conducted on an NVIDIA A8000 80GB GPU. The optimization was performed using the ADAM optimizer (Kingma & Ba, 2014) with an initial learning rate of $5 \times 10^{-4}$. A learning rate decay strategy was applied using the StepLR scheduler to facilitate gradual reduction during pre-training. During pre-training, we use $N = 10$ as the number of historical tokens, $K = 4$ as the number of prediction tokens, $P^* = 48$ as the reference patch size, and the batch size is set to 8192. Detailed configurations and parameter counts of the pre-trained models involved in this paper are provided in Table 10.

**Downstream Forecasting** In downstream forecasting, we configure the model to perform periodical patching based on the cycle length, tailored to the characteristics of each dataset. The number of historical tokens is set to $N = 10$, and the model is tasked with making predictions for target lengths of 96, 192, 336, and 720, respectively.

The *"Drop Last"* issue is reported by several researchers (Qiu et al., 2024; 2025a; Li et al., 2025). That is, in some previous works evaluating the model on test set with drop-last=True setting may cause additional errors related to test batch size. In our experiment, to ensure fair comparison in the future, we set the drop last to False for all baselines to avoid this issue.

*Table* 10. Detailed model configurations of LightGTS and corresponding parameter counts.

| Models | Encoder Layers | Decoder Layers | Model Dim. | FFN Dim. | Parameters |
|---|---|---|---|---|---|
| LightGTS-tiny | 1 | 1 | 256 | 512 | 1.3M |
| LightGTS-mini | 3 | 3 | 256 | 512 | 4M |

# B. More Results and Analysis

## B.1. Analysis of the Periods Finding

In this experiment, we investigate the effect of periodical finding methods. When prior knowledge is available or when there is sufficient data, we can treat the cycle length as known information. However, when prior knowledge is unavailable or data is insufficient, we can use the Fast Fourier Transform (FFT) to determine the cycle length based on the input sequence. As shown in Figure 6, the key observations are as follows:

- When the data exhibits strong periodicity (e.g., Solar, Electricity, Traffic), the cycle length extracted by FFT aligns well with the actual period, ensuring stable model performance.

- When the periodicity of the data is not pronounced (e.g., ETT, Weather), the cycle length extracted by FFT may not align with the intrinsic period, which can hinder model performance. However, LightGTS-FFT still outperforms the SOTA baselines.

- When the data exhibits little to no periodicity (e.g., Exchange), intrinsic period modeling is not crucial for the model's understanding of the time series. Therefore, regardless of the Periods Finding method used, its impact on model performance is minimal.

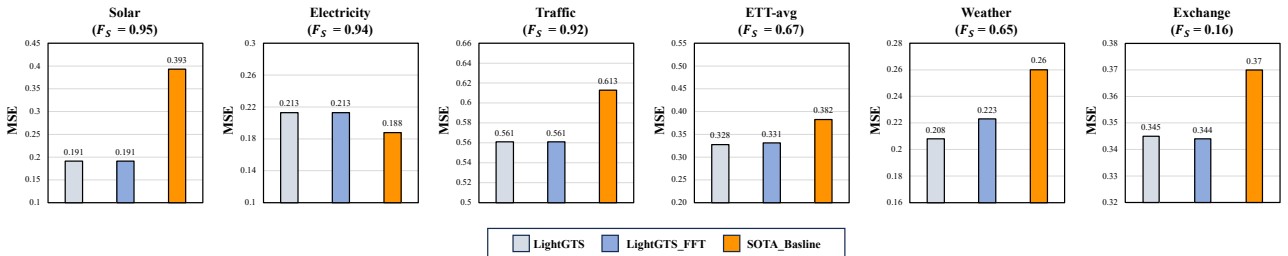

*Figure* 6. Comparisons of the model performance between different Periodical Finding methods in zero-shot setting. The SOTA baselines refer to the best-performing baseline results for each dataset. $F_S$ means the seasonality strength for each dataset.

## B.2. Generality of the Periodical Tokenization

Periodical tokenization can serve as a plug-in, adapting TSFMs that use patching methods (such as Timer) without the need for retraining. We applied periodical tokenization to Timer and tested its predictive performance in the zero-shot setting with different resizing methods, including the linear-resize and area-resize. The results, shown in Table 4, reveal that incorporating periodical tokenization significantly enhances performance across all resizing methods. Moreover, the flex-resize achieved a 19.23% improvement, outperforming the other two resizing methods by a significant margin.

*Table* 11. Ablation MSE results of the Flex-resize. All results are collected with periodical patching. The "Boost" indicates the percentage of performance improvement after incorporating the different resize techniques.

| Dataset | Weather | | | | Solar | | | |
|---|---|---|---|---|---|---|---|---|
| Horizon | 96 | 192 | 336 | 720 | 96 | 192 | 336 | 720 |
| Timer | 0.190 | 0.261 | 0.332 | 0.385 | 0.591 | 0.689 | 0.831 | 0.972 |
| + Linear-resize | 0.187 | 0.247 | 0.305 | 0.368 | 0.459 | 0.504 | 0.580 | 0.783 |
| Boost | 1.79% | 5.22% | 8.13% | 4.36% | 22.33% | 26.89% | 30.24% | 19.42% |
| + Area-resize | 0.186 | 0.246 | 0.304 | 0.367 | 0.461 | 0.506 | 0.581 | 0.782 |
| Boost | 2.23% | 5.65% | 8.41% | 4.62% | 21.93% | 26.62% | 30.08% | 19.57% |
| + Flex-resize | **0.178** | **0.238** | **0.297** | **0.357** | **0.425** | **0.488** | **0.558** | **0.671** |
| Boost | **6.24%** | **8.81%** | **10.54%** | **7.27%** | **28.05%** | **29.17%** | **32.85%** | **30.93%** |

# C. Full Experimental Results

## C.1. Zero-shot Forecasting

We provide all the results of the zero forecasting in Table 12. As shown in Table 10, we include nine representative real-world datasets, demonstrating that LightGTS achieves state-of-the-art forecasting performance.

## C.2. Full-shot Forecasting

Table 13 provides the comprehensive results for in-distribution forecasting, showcasing the performance of LightGTS-miny in both zero-shot and full-shot settings, as well as other baselines in the full-shot setting. Notably, LightGTS demonstrates superior performance over all baselines in the full-shot setting. Moreover, in the zero-shot setting, LightGTS achieves competitive results, rivaling the full-shot performance of other baseline models.

## C.3. Ablation Study

Table 14 presents the detailed results of the ablation studies for LightGTS in the zero-shot setting, including periodical tokenization and periodical parallel decoding.

Table 12. Full results of zero-shot forecasting experiments. Lower MSE or MAE values indicate better predictions. A dash ('-') denotes datasets included in the model's pretraining and therefore excluded from testing. **Red**: the best, Blue: the 2nd best.

| Models | LightGTS-tiny | | LightGTS-mini | | Timer | | MOIRAI | | Chronos | | TimesFM | | Time-MoE | |
|---|---|---|---|---|---|---|---|---|---|---|---|---|---|---|
| Metric | MSE | MAE | MSE | MAE | MSE | MAE | MSE | MAE | MSE | MAE | MSE | MAE | MSE | MAE |
| **ETTm1** 96 | 0.307 | 0.359 | **0.283** | **0.340** | 0.698 | 0.530 | 0.353 | 0.363 | 0.402 | 0.373 | 0.363 | 0.369 | 0.309 | 0.357 |
| 192 | 0.332 | 0.374 | 0.314 | 0.360 | 0.744 | 0.555 | 0.376 | 0.380 | 0.510 | 0.435 | 0.417 | 0.405 | 0.346 | 0.381 |
| 336 | 0.353 | 0.393 | **0.338** | **0.377** | 0.801 | 0.582 | 0.399 | 0.395 | 0.590 | 0.477 | 0.447 | 0.428 | 0.373 | 0.408 |
| 720 | 0.388 | **0.386** | **0.374** | 0.403 | 0.829 | 0.606 | 0.432 | 0.417 | 0.703 | 0.525 | 0.513 | 0.470 | 0.475 | 0.477 |
| avg | 0.345 | 0.378 | **0.327** | **0.370** | 0.768 | 0.568 | 0.390 | 0.389 | 0.551 | 0.453 | 0.435 | 0.418 | 0.376 | 0.406 |
| **ETTm2** 96 | 0.166 | 0.260 | **0.165** | **0.256** | 0.225 | 0.300 | 0.189 | 0.260 | 0.192 | 0.263 | 0.206 | 0.267 | 0.197 | 0.286 |
| 192 | 0.222 | 0.300 | **0.221** | **0.297** | 0.286 | 0.339 | 0.247 | 0.300 | 0.256 | 0.308 | 0.293 | 0.320 | 0.250 | 0.322 |
| 336 | **0.267** | 0.333 | 0.269 | **0.331** | 0.335 | 0.369 | 0.295 | 0.334 | 0.315 | 0.346 | 0.411 | 0.414 | 0.337 | 0.375 |
| 720 | 0.340 | **0.378** | **0.331** | 0.379 | 0.414 | 0.416 | 0.372 | 0.386 | 0.409 | 0.405 | 0.478 | 0.437 | 0.475 | 0.477 |
| avg | 0.249 | 0.318 | **0.247** | **0.316** | 0.315 | 0.356 | 0.276 | 0.320 | 0.293 | 0.331 | 0.347 | 0.360 | 0.315 | 0.365 |
| **ETTh1** 96 | 0.359 | 0.390 | **0.344** | **0.380** | 0.454 | 0.434 | 0.380 | 0.398 | 0.389 | 0.409 | 0.421 | 0.401 | 0.350 | 0.382 |
| 192 | 0.390 | 0.409 | **0.376** | **0.404** | 0.522 | 0.465 | 0.440 | 0.434 | 0.502 | 0.443 | 0.472 | 0.432 | 0.388 | 0.412 |
| 336 | 0.420 | 0.436 | **0.407** | 0.432 | 0.559 | 0.484 | 0.514 | 0.474 | 0.580 | 0.460 | 0.510 | 0.455 | 0.411 | **0.430** |
| 720 | 0.437 | 0.460 | **0.427** | 0.458 | 0.714 | 0.549 | 0.705 | 0.568 | 0.605 | 0.495 | 0.514 | 0.481 | 0.427 | **0.455** |
| avg | 0.401 | 0.424 | **0.388** | **0.419** | 0.562 | 0.483 | 0.510 | 0.469 | 0.519 | 0.452 | 0.479 | 0.442 | 0.394 | 0.420 |
| **ETTh2** 96 | **0.278** | 0.342 | 0.279 | 0.340 | 0.316 | 0.359 | 0.287 | **0.325** | 0.306 | 0.338 | 0.326 | 0.355 | 0.302 | 0.354 |
| 192 | 0.345 | 0.383 | **0.335** | 0.383 | 0.374 | 0.398 | 0.347 | **0.367** | 0.396 | 0.394 | 0.397 | 0.400 | 0.364 | 0.385 |
| 336 | 0.394 | 0.416 | **0.366** | 0.410 | 0.381 | 0.410 | 0.377 | **0.393** | 0.423 | 0.417 | 0.431 | 0.413 | 0.417 | 0.425 |
| 720 | 0.430 | 0.445 | 0.413 | 0.445 | 0.408 | 0.434 | **0.404** | **0.421** | 0.442 | 0.439 | 0.446 | 0.444 | 0.527 | 0.496 |
| avg | 0.362 | 0.397 | **0.348** | 0.395 | 0.370 | 0.400 | 0.354 | **0.377** | 0.392 | 0.397 | 0.400 | 0.403 | 0.403 | 0.415 |
| **Traffic** 96 | 0.525 | 0.356 | **0.463** | **0.331** | 0.526 | 0.368 | - | - | 0.562 | 0.378 | - | - | - | - |
| 192 | 0.547 | 0.365 | **0.504** | **0.353** | 0.561 | 0.385 | - | - | 0.579 | 0.412 | - | - | - | - |
| 336 | 0.645 | 0.419 | **0.593** | **0.401** | 0.614 | 0.412 | - | - | 0.594 | 0.420 | - | - | - | - |
| 720 | 0.722 | 0.457 | **0.685** | **0.440** | 0.749 | 0.464 | - | - | 0.723 | 0.472 | - | - | - | - |
| avg | 0.610 | 0.399 | **0.561** | **0.381** | 0.613 | 0.407 | - | - | 0.615 | 0.421 | - | - | - | - |
| **Weather** 96 | 0.151 | 0.211 | **0.141** | **0.195** | 0.190 | 0.236 | 0.177 | 0.208 | 0.186 | 0.208 | - | - | 0.159 | 0.213 |
| 192 | 0.191 | 0.248 | **0.180** | **0.235** | 0.261 | 0.293 | 0.219 | 0.249 | 0.238 | 0.258 | - | - | 0.215 | 0.266 |
| 336 | 0.236 | 0.284 | **0.224** | **0.272** | 0.332 | 0.340 | 0.277 | 0.292 | 0.313 | 0.353 | - | - | 0.291 | 0.322 |
| 720 | 0.299 | 0.323 | **0.286** | **0.320** | 0.385 | 0.381 | 0.365 | 0.350 | 0.416 | 0.415 | - | - | 0.415 | 0.400 |
| avg | 0.219 | 0.266 | **0.208** | **0.256** | 0.292 | 0.313 | 0.260 | 0.275 | 0.288 | 0.309 | - | - | 0.270 | 0.300 |
| **Exchange** 96 | **0.084** | **0.203** | 0.085 | 0.204 | 0.095 | 0.219 | 0.096 | 0.213 | 0.099 | 0.219 | 0.096 | 0.215 | 0.145 | 0.267 |
| 192 | **0.171** | **0.294** | 0.172 | 0.295 | 0.198 | 0.322 | 0.197 | 0.312 | 0.194 | 0.314 | 0.195 | 0.313 | 0.333 | 0.397 |
| 336 | **0.310** | **0.402** | 0.311 | 0.403 | 0.349 | 0.431 | 0.349 | 0.425 | 0.341 | 0.423 | 0.332 | 0.416 | 0.367 | 0.439 |
| 720 | **0.816** | **0.680** | 0.820 | 0.682 | 0.927 | 0.729 | 0.903 | 0.717 | 0.846 | 0.690 | 0.935 | 0.723 | 0.882 | 0.712 |
| avg | **0.345** | **0.395** | 0.347 | 0.396 | 0.392 | 0.425 | 0.386 | 0.417 | 0.370 | 0.412 | 0.390 | 0.417 | 0.432 | 0.454 |
| **Solar** 96 | 0.209 | 0.309 | **0.181** | **0.263** | 0.591 | 0.504 | 0.682 | 0.688 | 0.373 | 0.304 | 0.408 | 0.345 | 0.304 | 0.345 |
| 192 | 0.220 | 0.315 | **0.190** | **0.270** | 0.689 | 0.567 | 0.694 | 0.695 | 0.363 | 0.303 | 0.466 | 0.373 | 0.309 | 0.342 |
| 336 | 0.228 | 0.314 | **0.197** | **0.277** | 0.831 | 0.636 | 0.719 | 0.706 | 0.391 | 0.319 | 0.526 | 0.407 | 0.433 | 0.450 |
| 720 | 0.218 | 0.283 | **0.198** | **0.274** | 0.972 | 0.710 | 0.759 | 0.725 | 0.444 | 0.349 | 0.601 | 0.461 | 0.599 | 0.576 |
| avg | 0.219 | 0.305 | **0.191** | **0.271** | 0.771 | 0.604 | 0.714 | 0.704 | 0.393 | 0.319 | 0.500 | 0.397 | 0.411 | 0.428 |
| **Electricity** 96 | 0.181 | 0.272 | 0.156 | 0.255 | 0.210 | 0.312 | **0.152** | **0.242** | - | - | - | - | - | - |
| 192 | 0.195 | 0.284 | 0.183 | 0.282 | 0.239 | 0.337 | **0.171** | **0.259** | - | - | - | - | - | - |
| 336 | 0.253 | 0.338 | 0.240 | 0.338 | 0.284 | 0.372 | **0.192** | **0.278** | - | - | - | - | - | - |
| 720 | 0.304 | 0.382 | 0.275 | 0.357 | 0.456 | 0.479 | **0.236** | **0.313** | - | - | - | - | - | - |
| avg | 0.233 | 0.319 | 0.213 | 0.308 | 0.297 | 0.375 | **0.188** | **0.273** | - | - | - | - | - | - |
| 1st Count | 7 | 7 | **31** | **25** | 0 | 0 | 7 | 10 | 0 | 0 | 0 | 0 | 0 | 2 |

*Table* 13. The results of LightGTS-miny in zero-shot and full-shot setting and other baselines in full-shot setting. Lower MSE or MAE values indicate better predictions. **Red**: the best, Blue: the 2nd best.

| Models | | LightGTS (zero-shot) | | LightGTS (full-shot) | | PDF | | iTransformer | | Pathformer | | FITS | | TimeMixer | | PatchTST | |
|---|---|---|---|---|---|---|---|---|---|---|---|---|---|---|---|---|---|
| Metric | | MSE | MAE | MSE | MAE | MSE | MAE | MSE | MAE | MSE | MAE | MSE | MAE | MSE | MAE | MSE | MAE |
| ETTm1 | 96 | 0.283 | 0.340 | **0.266** | **0.320** | 0.286 | 0.340 | 0.287 | 0.342 | 0.290 | 0.335 | 0.303 | 0.345 | 0.293 | 0.345 | 0.289 | 0.343 |
| | 192 | 0.314 | 0.360 | **0.307** | **0.349** | 0.321 | 0.364 | 0.331 | 0.371 | 0.337 | 0.363 | 0.337 | 0.365 | 0.335 | 0.372 | 0.329 | 0.368 |
| | 336 | 0.338 | 0.377 | **0.334** | **0.370** | 0.354 | 0.383 | 0.358 | 0.384 | 0.374 | 0.384 | 0.368 | 0.384 | 0.368 | 0.386 | 0.362 | 0.390 |
| | 720 | **0.374** | **0.403** | 0.377 | 0.405 | 0.408 | 0.415 | 0.412 | 0.416 | 0.428 | 0.416 | 0.420 | 0.413 | 0.426 | 0.417 | 0.416 | 0.423 |
| | avg | 0.327 | 0.370 | **0.321** | **0.361** | 0.342 | 0.376 | 0.347 | 0.378 | 0.357 | 0.375 | 0.357 | 0.377 | 0.356 | 0.380 | 0.349 | 0.381 |
| ETTm2 | 96 | 0.165 | 0.256 | **0.153** | **0.241** | 0.163 | 0.251 | 0.168 | 0.262 | 0.164 | 0.250 | 0.165 | 0.254 | 0.165 | 0.256 | 0.165 | 0.255 |
| | 192 | 0.221 | 0.297 | **0.209** | **0.286** | 0.219 | 0.290 | 0.224 | 0.295 | 0.219 | 0.288 | 0.219 | 0.291 | 0.225 | 0.298 | 0.221 | 0.293 |
| | 336 | 0.269 | 0.331 | **0.259** | **0.318** | 0.269 | 0.330 | 0.274 | 0.330 | 0.267 | 0.319 | 0.272 | 0.326 | 0.277 | 0.332 | 0.276 | 0.327 |
| | 720 | **0.331** | 0.379 | 0.336 | **0.367** | 0.349 | 0.382 | 0.367 | 0.385 | 0.361 | 0.377 | 0.359 | 0.381 | 0.360 | 0.387 | 0.362 | 0.381 |
| | avg | 0.247 | 0.316 | **0.239** | **0.303** | 0.250 | 0.313 | 0.258 | 0.318 | 0.253 | 0.309 | 0.254 | 0.313 | 0.257 | 0.318 | 0.256 | 0.314 |
| ETTh1 | 96 | 0.344 | 0.380 | **0.340** | **0.375** | 0.360 | 0.391 | 0.386 | 0.405 | 0.372 | 0.392 | 0.376 | 0.396 | 0.372 | 0.401 | 0.377 | 0.397 |
| | 192 | 0.376 | 0.404 | **0.373** | **0.401** | 0.392 | 0.414 | 0.430 | 0.435 | 0.408 | 0.415 | 0.400 | 0.418 | 0.413 | 0.430 | 0.409 | 0.425 |
| | 336 | 0.407 | 0.432 | **0.404** | **0.420** | 0.418 | 0.435 | 0.450 | 0.452 | 0.438 | 0.434 | 0.419 | 0.435 | 0.438 | 0.450 | 0.431 | 0.444 |
| | 720 | **0.427** | 0.458 | 0.435 | **0.456** | 0.456 | 0.462 | 0.495 | 0.487 | 0.450 | 0.463 | 0.435 | 0.458 | 0.483 | 0.483 | 0.457 | 0.477 |
| | avg | 0.388 | 0.419 | **0.388** | **0.413** | 0.407 | 0.426 | 0.440 | 0.445 | 0.417 | 0.426 | 0.408 | 0.427 | 0.427 | 0.441 | 0.419 | 0.436 |
| ETTh2 | 96 | 0.279 | 0.340 | **0.268** | **0.326** | 0.276 | 0.341 | 0.292 | 0.347 | 0.279 | 0.336 | 0.277 | 0.345 | 0.270 | 0.342 | 0.274 | 0.337 |
| | 192 | 0.335 | 0.383 | 0.333 | **0.369** | 0.339 | 0.382 | 0.348 | 0.384 | 0.345 | 0.380 | 0.331 | 0.379 | 0.349 | 0.387 | 0.348 | 0.384 |
| | 336 | 0.366 | 0.410 | 0.353 | **0.390** | 0.374 | 0.406 | 0.372 | 0.407 | 0.378 | 0.408 | 0.350 | 0.396 | 0.367 | 0.410 | 0.377 | 0.416 |
| | 720 | 0.413 | 0.445 | 0.385 | **0.425** | 0.398 | 0.433 | 0.424 | 0.444 | 0.437 | 0.455 | 0.382 | 0.425 | 0.401 | 0.436 | 0.406 | 0.441 |
| | avg | 0.348 | 0.395 | **0.335** | **0.377** | 0.347 | 0.391 | 0.359 | 0.396 | 0.360 | 0.395 | 0.335 | 0.386 | 0.347 | 0.394 | 0.351 | 0.395 |
| Traffic | 96 | 0.463 | 0.331 | **0.359** | **0.244** | 0.368 | 0.252 | 0.363 | 0.265 | 0.384 | 0.250 | 0.400 | 0.280 | 0.369 | 0.256 | 0.370 | 0.262 |
| | 192 | 0.504 | 0.353 | 0.382 | **0.250** | 0.382 | 0.261 | 0.384 | 0.273 | 0.405 | 0.257 | 0.412 | 0.288 | 0.400 | 0.271 | 0.386 | 0.269 |
| | 336 | 0.593 | 0.401 | 0.395 | **0.262** | **0.393** | 0.268 | 0.396 | 0.277 | 0.424 | 0.265 | 0.426 | 0.301 | 0.407 | 0.272 | 0.396 | 0.275 |
| | 720 | 0.685 | 0.440 | 0.435 | **0.279** | 0.438 | 0.297 | 0.445 | 0.308 | 0.452 | 0.283 | 0.478 | 0.339 | 0.462 | 0.316 | 0.435 | 0.295 |
| | avg | 0.561 | 0.381 | **0.393** | **0.259** | 0.395 | 0.270 | 0.397 | 0.281 | 0.416 | 0.264 | 0.429 | 0.302 | 0.410 | 0.279 | 0.397 | 0.275 |
| Weather | 96 | 0.141 | 0.195 | **0.139** | **0.182** | 0.147 | 0.196 | 0.157 | 0.207 | 0.148 | 0.195 | 0.172 | 0.225 | 0.147 | 0.198 | 0.149 | 0.196 |
| | 192 | **0.180** | 0.235 | 0.180 | **0.224** | 0.193 | 0.240 | 0.200 | 0.248 | 0.191 | 0.235 | 0.215 | 0.261 | 0.191 | 0.242 | 0.191 | 0.239 |
| | 336 | **0.224** | 0.272 | 0.225 | **0.262** | 0.245 | 0.280 | 0.252 | 0.287 | 0.243 | 0.274 | 0.261 | 0.295 | 0.244 | 0.280 | 0.242 | 0.279 |
| | 720 | 0.286 | 0.320 | **0.284** | **0.309** | 0.323 | 0.334 | 0.320 | 0.336 | 0.318 | 0.326 | 0.326 | 0.341 | 0.316 | 0.331 | 0.312 | 0.330 |
| | avg | 0.208 | 0.256 | **0.207** | **0.244** | 0.227 | 0.263 | 0.232 | 0.270 | 0.225 | 0.258 | 0.244 | 0.281 | 0.225 | 0.263 | 0.224 | 0.261 |
| Exchange | 96 | 0.085 | 0.204 | 0.079 | **0.197** | 0.083 | 0.200 | 0.080 | 0.201 | 0.088 | 0.208 | 0.082 | 0.199 | 0.087 | 0.209 | 0.079 | 0.200 |
| | 192 | 0.172 | 0.295 | 0.162 | 0.290 | 0.172 | 0.294 | **0.155** | **0.287** | 0.183 | 0.304 | 0.173 | 0.295 | 0.178 | 0.300 | 0.159 | 0.289 |
| | 336 | 0.311 | 0.403 | **0.295** | **0.395** | 0.323 | 0.411 | 0.298 | 0.399 | 0.354 | 0.429 | 0.317 | 0.406 | 0.376 | 0.451 | 0.297 | 0.399 |
| | 720 | 0.820 | 0.682 | 0.750 | 0.650 | 0.820 | 0.682 | **0.749** | 0.650 | 0.909 | 0.716 | 0.825 | 0.684 | 0.897 | 0.711 | 0.751 | 0.650 |
| | avg | 0.347 | 0.396 | 0.322 | **0.383** | 0.350 | 0.397 | **0.321** | 0.384 | 0.384 | 0.414 | 0.349 | 0.396 | 0.385 | 0.418 | 0.322 | 0.385 |
| Solar | 96 | 0.181 | 0.263 | **0.160** | **0.206** | 0.181 | 0.247 | 0.174 | 0.229 | 0.218 | 0.235 | 0.208 | 0.255 | 0.180 | 0.233 | 0.170 | 0.234 |
| | 192 | 0.190 | 0.270 | **0.180** | **0.219** | 0.200 | 0.259 | 0.205 | 0.270 | 0.196 | 0.220 | 0.229 | 0.267 | 0.201 | 0.259 | 0.204 | 0.302 |
| | 336 | 0.197 | 0.277 | **0.188** | 0.220 | 0.208 | 0.269 | 0.216 | 0.282 | 0.195 | 0.220 | 0.241 | 0.273 | 0.214 | 0.272 | 0.212 | 0.293 |
| | 720 | 0.198 | 0.274 | **0.190** | **0.234** | 0.212 | 0.275 | 0.211 | 0.260 | 0.208 | 0.237 | 0.248 | 0.277 | 0.218 | 0.278 | 0.215 | 0.307 |
| | avg | 0.191 | 0.271 | **0.179** | **0.220** | 0.200 | 0.263 | 0.202 | 0.260 | 0.204 | 0.228 | 0.232 | 0.268 | 0.203 | 0.261 | 0.200 | 0.284 |
| Electricity | 96 | 0.156 | 0.255 | **0.124** | **0.216** | 0.128 | 0.222 | 0.134 | 0.230 | 0.135 | 0.222 | 0.139 | 0.237 | 0.153 | 0.256 | 0.143 | 0.247 |
| | 192 | 0.183 | 0.282 | **0.145** | **0.240** | 0.147 | 0.242 | 0.154 | 0.250 | 0.157 | 0.253 | 0.154 | 0.250 | 0.168 | 0.269 | 0.158 | 0.260 |
| | 336 | 0.240 | 0.338 | **0.163** | **0.252** | 0.165 | 0.260 | 0.169 | 0.265 | 0.170 | 0.267 | 0.170 | 0.268 | 0.189 | 0.291 | 0.168 | 0.267 |
| | 720 | 0.275 | 0.357 | **0.191** | **0.282** | 0.199 | 0.289 | 0.194 | 0.288 | 0.211 | 0.302 | 0.212 | 0.304 | 0.228 | 0.320 | 0.214 | 0.307 |
| | avg | 0.213 | 0.308 | **0.156** | **0.248** | 0.160 | 0.253 | 0.163 | 0.258 | 0.168 | 0.261 | 0.169 | 0.265 | 0.185 | 0.284 | 0.171 | 0.270 |
| 1st Count | | 5 | 1 | **31** | **42** | 0 | 0 | 3 | 1 | 0 | 0 | 3 | 0 | 0 | 0 | 0 | 0 |

*Table* 14. Ablations on key components of LightGTS in zero-shot setting, including periodical tokenization, and periodical parallel decoing.

| Decoding | Periodical Parallel Decoding | | | | Autoregressive Decoding | | | | Masked Autoencoder | | | |
|---|---|---|---|---|---|---|---|---|---|---|---|---|
| Tokenization | Periodical | | Fixed | | Periodical | | Fixed | | Periodical | | Fixed | |
| Metric | MSE | MAE | MSE | MAE | MSE | MAE | MSE | MAE | MSE | MAE | MSE | MAE |
| ETTm1 96 | 0.283 | 0.340 | 0.658 | 0.509 | 0.320 | 0.368 | 0.623 | 0.499 | 0.351 | 0.385 | 0.768 | 0.572 |
| ETTm1 192 | 0.314 | 0.360 | 0.667 | 0.523 | 0.342 | 0.381 | 0.659 | 0.515 | 0.371 | 0.397 | 0.792 | 0.593 |
| ETTm1 336 | 0.338 | 0.377 | 0.686 | 0.541 | 0.361 | 0.394 | 0.704 | 0.544 | 0.387 | 0.411 | 0.808 | 0.603 |
| ETTm1 720 | 0.374 | 0.403 | 0.704 | 0.559 | 0.390 | 0.414 | 0.734 | 0.565 | 0.448 | 0.457 | 0.852 | 0.627 |
| ETTm1 avg | **0.327** | **0.370** | 0.679 | 0.533 | 0.353 | 0.389 | 0.680 | 0.531 | 0.389 | 0.412 | 0.805 | 0.599 |
| ETTm2 96 | 0.165 | 0.256 | 0.210 | 0.297 | 0.169 | 0.260 | 0.208 | 0.294 | 0.202 | 0.291 | 0.308 | 0.356 |
| ETTm2 192 | 0.221 | 0.297 | 0.276 | 0.335 | 0.226 | 0.300 | 0.280 | 0.339 | 0.259 | 0.328 | 0.391 | 0.404 |
| ETTm2 336 | 0.269 | 0.331 | 0.336 | 0.369 | 0.272 | 0.335 | 0.346 | 0.380 | 0.293 | 0.355 | 0.423 | 0.425 |
| ETTm2 720 | 0.331 | 0.379 | 0.433 | 0.423 | 0.331 | 0.380 | 0.444 | 0.436 | 0.379 | 0.420 | 0.492 | 0.466 |
| ETTm2 avg | **0.247** | **0.316** | 0.314 | 0.356 | 0.250 | 0.319 | 0.320 | 0.362 | 0.283 | 0.349 | 0.404 | 0.413 |
| ETTh1 96 | 0.344 | 0.380 | 0.364 | 0.396 | 0.365 | 0.397 | 0.365 | 0.397 | 0.401 | 0.423 | 0.393 | 0.415 |
| ETTh1 192 | 0.376 | 0.404 | 0.391 | 0.413 | 0.392 | 0.415 | 0.392 | 0.415 | 0.433 | 0.449 | 0.452 | 0.459 |
| ETTh1 336 | 0.407 | 0.432 | 0.409 | 0.430 | 0.401 | 0.427 | 0.411 | 0.436 | 0.444 | 0.463 | 0.532 | 0.513 |
| ETTh1 720 | 0.427 | 0.458 | 0.421 | 0.451 | 0.420 | 0.453 | 0.432 | 0.453 | 0.534 | 0.529 | 0.623 | 0.572 |
| ETTh1 avg | **0.388** | **0.419** | 0.396 | 0.423 | 0.394 | 0.423 | 0.400 | 0.425 | 0.453 | 0.466 | 0.500 | 0.490 |
| ETTh2 96 | 0.279 | 0.340 | 0.283 | 0.346 | 0.279 | 0.341 | 0.285 | 0.344 | 0.307 | 0.368 | 0.378 | 0.411 |
| ETTh2 192 | 0.335 | 0.383 | 0.339 | 0.385 | 0.339 | 0.384 | 0.342 | 0.387 | 0.383 | 0.423 | 0.454 | 0.461 |
| ETTh2 336 | 0.366 | 0.410 | 0.382 | 0.413 | 0.386 | 0.419 | 0.385 | 0.419 | 0.464 | 0.463 | 0.463 | 0.470 |
| ETTh2 720 | 0.413 | 0.445 | 0.412 | 0.438 | 0.463 | 0.469 | 0.455 | 0.465 | 0.550 | 0.517 | 0.465 | 0.477 |
| ETTh2 avg | **0.348** | **0.395** | 0.354 | 0.396 | 0.367 | 0.403 | 0.367 | 0.404 | 0.426 | 0.443 | 0.440 | 0.455 |
| Weather 96 | 0.141 | 0.195 | 0.164 | 0.210 | 0.154 | 0.211 | 0.164 | 0.211 | 0.188 | 0.249 | 0.239 | 0.281 |
| Weather 192 | 0.180 | 0.235 | 0.219 | 0.261 | 0.194 | 0.248 | 0.219 | 0.266 | 0.227 | 0.278 | 0.319 | 0.333 |
| Weather 336 | 0.224 | 0.272 | 0.282 | 0.308 | 0.244 | 0.286 | 0.285 | 0.315 | 0.273 | 0.309 | 0.359 | 0.363 |
| Weather 720 | 0.286 | 0.320 | 0.381 | 0.371 | 0.314 | 0.336 | 0.392 | 0.379 | 0.351 | 0.368 | 0.437 | 0.417 |
| Weather avg | **0.208** | **0.256** | 0.262 | 0.288 | 0.226 | 0.270 | 0.265 | 0.293 | 0.260 | 0.301 | 0.339 | 0.349 |
| Electricity 96 | 0.156 | 0.255 | 0.192 | 0.287 | 0.197 | 0.287 | 0.199 | 0.297 | 0.245 | 0.321 | 0.207 | 0.299 |
| Electricity 192 | 0.183 | 0.282 | 0.205 | 0.297 | 0.210 | 0.303 | 0.212 | 0.307 | 0.277 | 0.355 | 0.263 | 0.357 |
| Electricity 336 | 0.240 | 0.338 | 0.231 | 0.320 | 0.233 | 0.326 | 0.236 | 0.327 | 0.318 | 0.395 | 0.396 | 0.460 |
| Electricity 720 | 0.275 | 0.357 | 0.275 | 0.355 | 0.276 | 0.361 | 0.279 | 0.372 | 0.450 | 0.495 | 0.623 | 0.597 |
| Electricity avg | **0.213** | **0.308** | 0.226 | 0.315 | 0.229 | 0.319 | 0.231 | 0.326 | 0.322 | 0.392 | 0.372 | 0.428 |
| Traffic 96 | 0.463 | 0.331 | 0.579 | 0.383 | 0.603 | 0.392 | 0.592 | 0.394 | 0.656 | 0.425 | 0.607 | 0.402 |
| Traffic 192 | 0.504 | 0.353 | 0.591 | 0.389 | 0.612 | 0.402 | 0.600 | 0.396 | 0.690 | 0.453 | 0.686 | 0.470 |
| Traffic 336 | 0.593 | 0.401 | 0.630 | 0.407 | 0.639 | 0.408 | 0.641 | 0.417 | 0.740 | 0.487 | 0.874 | 0.586 |
| Traffic 720 | 0.685 | 0.440 | 0.686 | 0.433 | 0.682 | 0.440 | 0.686 | 0.438 | 0.900 | 0.571 | 1.044 | 0.679 |
| Traffic avg | **0.561** | **0.381** | 0.621 | 0.403 | 0.634 | 0.410 | 0.630 | 0.411 | 0.746 | 0.484 | 0.803 | 0.534 |
| Solar 96 | 0.181 | 0.263 | 0.222 | 0.304 | 0.190 | 0.270 | 0.212 | 0.308 | 0.205 | 0.290 | 0.464 | 0.490 |
| Solar 192 | 0.190 | 0.270 | 0.345 | 0.371 | 0.207 | 0.282 | 0.368 | 0.385 | 0.213 | 0.294 | 0.508 | 0.504 |
| Solar 336 | 0.197 | 0.277 | 0.400 | 0.423 | 0.211 | 0.287 | 0.446 | 0.468 | 0.217 | 0.300 | 0.561 | 0.532 |
| Solar 720 | 0.198 | 0.274 | 0.529 | 0.542 | 0.215 | 0.296 | 0.535 | 0.505 | 0.235 | 0.330 | 0.724 | 0.645 |
| Solar avg | **0.191** | **0.271** | 0.374 | 0.410 | 0.206 | 0.284 | 0.390 | 0.417 | 0.217 | 0.304 | 0.564 | 0.543 |

