# OpenReview forum: "LightGTS: A Lightweight General Time Series Forecasting Model"
_ICML.cc/2025/Conference — ICML 2025 poster_

### Official Review · Reviewer_Kj5L · 2025-03-06

**Overall Recommendation:** 3

**Summary:**

This paper presents LightGTS, a lightweight base model for time-series forecasting, along with Periodical Patch Embedding to adapt to intrinsic periodic differences across datasets and Periodical Parallel Decoding to prevent error accumulation. LightGTS achieves 10 to 100 times size reduction based only on 4 million parameters.

**Claims And Evidence:**

yes

**Essential References Not Discussed:**

n/a

**Experimental Designs Or Analyses:**

should add more similar works into comparsion

**Methods And Evaluation Criteria:**

yes

**Other Comments Or Suggestions:**

n/a

**Other Strengths And Weaknesses:**

pros:

1. The proposed techniques are plug-and-play and can be combined with many time series forecasting models.

2.The model achieves very good performance despite its extremely lightweight design.

cons:

1. Although the idea of training a lightweight temporal foundation model is quite interesting, there have been some cutting-edge works that made similar attempts, such as Tiny Time Mixers (TTMs): Fast Pre-trained Models for Enhanced Zero/Few-Shot Forecasting of Multivariate Time Series. Moreover, the significant contribution of this paper is very similar to that of the article, which also puts forward an Adaptive patching technology. I think the authors should have more discussions with relevant literature and add corresponding experimental comparisons.

2. The article's expression and description need some revision. Last word of the first sentence of the abstract should perhaps be 'pretraining' instead of 'pertaining'. It might be helpful to include a diagram illustrating the differences between the Periodical Patching and Periodical Parallel Decoding techniques and traditional practices. This will better help readers understand.

**Questions For Authors:**

see concerns above

**Relation To Broader Scientific Literature:**

the significant contribution of this paper is very similar to that of the article, which also puts forward an Adaptive patching technology. such as Tiny Time Mixers (TTMs)

**Theoretical Claims:**

yes

---

> ### Author Rebuttal · Authors · 2025-04-01
>
> **Q1: Discussions and experimental comparisons with Tiny Time Mixers (TTMs).**
>
> A1: Thank you for mentioning TTMs, which are also lightweight TSFMs like LightGTS. We will differentiate LightGTS from TTMs in the following two aspects:
>
> - **Flexibility**: TTMs have fixed input and output formats, which imposes limitations in downstream applications. In contrast, LightGTS supports flexible input and output configurations.
> - **Adaptive Patching**: While TTMs employ adaptive patching through CV-inspired patch merging techniques to capture multi-scale features, they remain constrained by predefined patch sizes. LightGTS, however, leverages periodical patching that adaptively segments time series based on the intrinsic periods. This approach enables LightGTS to achieve unified modeling across datasets with varying scales.
>
> Based on your suggestions, we add a comparison experiment with TTMs. The following table shows the average MSE in zero shot setting, and LightGTS outperforms TTMs on most datasets.
>
> | Model | ETTh1 | ETTh2 | ETTm1 | ETTm2 | weather | electricity |
> | --- | --- | --- | --- | --- | --- | --- |
> | LightGTS-mini | **0.388** | 0.348 | **0.327** | **0.247** | **0.208** | 0.213 |
> | TTMs-Advanced | 0.4 | **0.333** | 0.362 | 0.252 | 0.231 | **0.192** |
>
> **Q2: The article's expression and description need some revision.**
>
> A2: Thank you for your detailed suggestions, we will correct the errors you mentioned in the revised paper.
>
> **Q3: Need a diagram illustrating the differences between the Periodical Patching and Periodical Parallel Decoding techniques and traditional practices.**
>
> A3:
>
> - As shown in Figure 2 (a), the key difference of Periodical Patching from other patching techniques is its ability to adaptively perform consistent modeling on multivariate time series datasets with varying scales.
> - The differences between Periodical Parallel Decoding and other decoding techniques are shown in the table below.
>
>
> | Decoding techniques | Flexible Input/Output | Hight Inference | Pre-train/Downstream Consistency |
> | --- | --- | --- | --- |
> | Auto-regressive Decoding | √ | X | √ |
> | MLP Decoding | X | √ | √ |
> | Masked Auto-encoders | √ | √ | X |
> | Periodical Parallel Decoding | √ | √ | √ |

---

### Official Review · Reviewer_CW8j · 2025-03-11

**Overall Recommendation:** 3

**Summary:**

The paper introduces LightGTS, a lightweight time series forecasting model leveraging consistent periodical modeling. It proposes a periodical tokenization, which adaptively splits time series into patches aligned with intrinsic periods to handle varying scales, and periodical parallel decoding, which leverages historical tokens to improve forecasting while avoiding autoregressive errors. Experments show that LightGTS achieves SOTA performance on 9 benchmarks in zero/full-shot settings with <5M parameters, significantly improving efficiency over existing foundation models.

**Claims And Evidence:**

The claims regarding improved generalization and handling diverse scales through periodic tokenization are supported by case studies (e.g., Figure 2b) and multi-dataset benchmark experiments. However, the assumption of 'consistent periodic patterns' relies on accurately detecting intrinsic periods, yet the validation of this detection remains unclear. While the authors provide a comparison in Appendix Figure 6, the distinction between LightGTS and LightGTS-FFT is not well explained. Additionally, results indicate that the proposed method underperforms on strongly periodic data (e.g., Electricity) but excels when periodicity is less pronounced. This contradicts the claims and explanations in Appendix B.1.

**Essential References Not Discussed:**

N/A

**Experimental Designs Or Analyses:**

The experiments span diverse datasets with a well-reasoned design, including zero/full-shot settings and hyper-parameter studies. However, the zero-shot evaluation requires further validation for generalizability to unseen scales and periods, such as an ablation study on cycle-length estimation.

**Methods And Evaluation Criteria:**

Periodical tokenization aligns with time series' cyclical inductive bias, addressing fixed tokenization's limitations. Parallel decoding avoids error accumulation, a known issue in autoregressive models. Evaluation on 9 benchmarks is reasonable.

**Other Comments Or Suggestions:**

Clarify the detection and effectiveness of intrinsic periods across datasets.


## update after rebuttal
The concerns raised have been addressed. I will maintain the original acceptance score.

**Other Strengths And Weaknesses:**

Strengths: Novel integration of periodicity into tokenization/decoding, strong empirical results, and practical efficiency.

Weaknesses: Cycle-length detection ablation is unclear; limited discussion of failure modes (e.g., non-periodic series).

**Questions For Authors:**

Cycle detection: How are intrinsic periods determined for datasets with unknown/noisy periodicities?

**Relation To Broader Scientific Literature:**

LightGTS builds on time series foundation models by addressing fixed tokenization limitations. It aligns with recent trends in adaptive tokenization but uniquely integrates periodicity. The decoding approach relates to non-autoregressive methods in NLP/TSF, though periodicity integration is novel.

**Theoretical Claims:**

The author provides a brief theoretical analysis in Sec. 3.2, focusing on explaining the resize transformation of the projection layer weights.

---

> ### Author Rebuttal · Authors · 2025-04-01
>
> **Q1: The zero-shot evaluation requires further validation for generalizability to unseen scales and periods, such as an ablation study on cycle-length estimation.**
>
> A1: To further evaluate the generalizability of LightGTS, we add experiments on Chronos Benchmark II which contains 27 evaluation datasets with different scales and periods. As shown in the table below, LightGTS shows outstanding performance, second only to Moirai-large whose training corpus overlap is much higher. In addition, LightGTS also exhibits superior efficiency compared to other foundation model.
>
> | Model | Moirai-large | Chronos-large | TimesFM | LightGTS_mini(ours) | Seasonal Naive |
> | --- | --- | --- | --- | --- | --- |
> | Average Relative Error | **0.791** | 0.824 | 0.882 | 0.819 | 1.000 |
> | Median inference time (s) | 14.254 | 26.827 | 0.396 | 1.390 | **0.096** |
> | Training corpus overlap (%) | 81.5% | 0% | 14.8% | 31% | 0% |
>
> **Q2: Cycle-length detection ablation is unclear; limited discussion of failure modes (e.g., non-periodic series).**
>
> A2: Actually, for datasets with little periodicity, LightGTS is insensitive to the prediction of trends with the selection of patch size. Take the exchange dataset which exhibits little periodicity as an example, the performance of LightGTS in zero-shot setting remains stable when the patch size is set to 1, 4, 6, 8, 16 and 24, respectively. In the paper, we use fft-extracted period: 6.
>
> | Patch size | 1 | 4 | 6 (fft-extracted) | 8 | 16 | 24 |
> | --- | --- | --- | --- | --- | --- | --- |
> | Exchange-MSE | 0.347 | 0.348 | 0.347 | 0.353 | 0.351 | 0.354 |
>
> **Q3: Clarify the detection and effectiveness of intrinsic periods across datasets. The periods determined for datasets with unknown/noisy periodicities?**
>
> A3:  For datasets with prior knowledge of period, we use known period. For datasets where the period is not known, we use Fast Fourier Transform (FFT) to extract the period, which we have discussed in the Appendix B.1.
>
> In Table 8, we give the cycle lengths of the evaluation dataset. For traffic and electricity dataset with two cycle lengths: 24 (one day) and 168 (one week), we choose the smaller cycle length 24 as the patch size. For the exchange dataset with insignificant periodicity, we use fft to extract the cycle length of 6.
>
> In Table 4 of  paper, we perform an ablation study against periodical patching on seven datasets. After replacing periodical patching with a fixed patching, the model’s performance decreases significantly, validating the effectiveness of periodical patching.

---

> > ### Comment · Reviewer_CW8j · 2025-04-08
> >
> > The concerns raised have been addressed. I will maintain the original acceptance score.

---

### Official Review · Reviewer_AuKx · 2025-03-11

**Overall Recommendation:** 3

**Summary:**

In this paper, the authors proposed a lightweight pretrained TSF model with a new tokenization technique. With the proposed periodical tokenization method, the authors claimed that one can naturally deal with time series with different granularity and periodicity. In addition, it significantly reduced the number of model parameters for pretraining, while achieving SOTA performance on one benchmark dataset.

**Claims And Evidence:**

No, the authors claimed that they proposed a general time series forecasting model. However, the proposed periodical patching and periodical parallel decoding, the core component of the proposed LightGTS, are only designed for periodical time series data. In addition, they only evaluate datasets with strong periodic (long-term TSF benchmark). Therefore, it is not clear how the proposed method performs on non-periodic time series, which are prevalent in the real world as well. Therefore, in my opinion, it is not appropriate to claim the statement of the general TSF model.

**Essential References Not Discussed:**

Light-weight time series models [1] [2] or even non-parametric models [3] that use periodical information to improve forecasting should be discussed.
[1] Lin, S., Lin, W., Wu, W., Chen, H., & Yang, J. (2024). Sparsetsf: Modeling long-term time series forecasting with 1k parameters. arXiv preprint arXiv:2405.00946.
[2] Lin, S., Lin, W., Hu, X., Wu, W., Mo, R., & Zhong, H. (2024). Cyclenet: enhancing time series forecasting through modeling periodic patterns. Advances in Neural Information Processing Systems, 37, 106315-106345.
[3] He, X. Li, Y., Tan, J., Wu B. & Li, F. (2023).  OneShotSTL: One-Shot Seasonal-Trend Decomposition For Online Time Series Anomaly Detection And Forecasting. VLDB Endowment, 16, 06, 1399-1412.

**Experimental Designs Or Analyses:**

The choice of evaluation datasets should be improved.
The experimental settings are incomplete.

**Methods And Evaluation Criteria:**

They only evaluated datasets with strong periodic (long-term TSF benchmark). Larger benchmark datasets should be used for evaluating the pretrained TSF model, e.g., GIFT-Eval.

**Other Comments Or Suggestions:**

Page 1, line 12, pertaining -> pretraining

**Other Strengths And Weaknesses:**

Strengths:
1. The proposed periodical patching and projection layer looks novel and interesting. It seems to be a good way to handle datasets with different gradualility and periodicity.

Weeknesses
1. How the method would handle periodic and non-periodic datasets together is not clear.
2. Again, the evaluation datasets are not enough.
3. Specific experimental settings should be given. For instance, which period is used for training and evaluation? How long is the context length?
4. Some related works are missing.

**Questions For Authors:**

1. How would the proposed model perform trend forecasting? Does it go back to point patch? Then the model size would be significantly affected, if there are many non-periodic time series in the pretraining datasets, right? Emperically, how does the proposed model perform on such datasets?
2. How would the proposed method deal with different cycle lengths? For instance, there are daily and weekly periods in traffic data. How did the authors pick them?

**Relation To Broader Scientific Literature:**

The proposed a new tokenization method for time series datasets utilizing periodical information. The idea seems to be novel and technically sounds.

**Theoretical Claims:**

No, there are no theoretical claims in the paper.

---

> ### Author Rebuttal · Authors · 2025-04-01
>
> **Q1: Larger benchmark datasets should be used for evaluating the pretrained TSF model.**
>
> A1: We use Chronos Benchmark II to further evaluate the effectiveness of LightGTS. As shown in the table below, LightGTS shows outstanding performance, second only to Moirai-large whose training corpus overlap is much higher. In addition, LightGTS also exhibits superior efficiency compared to other foundation model.
>
> | Model | Moirai-large | Chronos-large | TimesFM | LightGTS_mini(ours) | Seasonal Naive |
> | --- | --- | --- | --- | --- | --- |
> | Average Relative Error | **0.791** | 0.824 | 0.882 | 0.819 | 1.000 |
> | Median inference time (s) | 14.254 | 26.827 | 0.396 | 1.390 | **0.096** |
> | Training corpus overlap (%) | 81.5% | 0% | 14.8% | 31% | 0% |
>
> **Q2: Light-weight time series models [1] [2] or even non-parametric models [3] that use periodical information to improve forecasting should be discussed.**
>
> A2:
>
> - CycleNet, SparseTSF, and OneShotSTL all propose explicit periodicity decomposition methods to enhance the accuracy and lightweight nature of time series forecasting models. However, LightGTS not only explicitly models periodicity through periodic patching but also adapts to multivariate time series pretraining via flex resizing technique. This allows the model to leverage the scale-invariant periodicity inductive bias in multivariate time series, achieving strong zero-shot performance with fewer parameters.
> - Experiment (MSE)：
>
> | Datasets | LightGTS_mini | SparseTSF | CycleNet | OneShotSTL |
> | --- | --- | --- | --- | --- |
> | ETTm2 | **0.239** | 0.251 | 0.244 | 0.255 |
> | Electricity | **0.156** | 0.162 | 0.159 | 0.167 |
> | Traffic | **0.393** | 0.404 | 0.397 | 0.403 |
>
> **Q3: Specific experimental settings should be given. For instance, which period is used for training and evaluation? How long is the context length?**
>
> A3:
>
> - For datasets with prior knowledge of period, we use known period. For datasets where the period is not known, we use Fast Fourier Transform (FFT) to extract the period.
> - We use a variable context length, keeping the number of tokens at 10. For example, for a dataset with a period of 24, we set the context length to 240.
>
> **Q4: How would the proposed model perform trend forecasting? Does it go back to point patch? Then the model size would be significantly affected, if there are many non-periodic time series in the pretraining datasets, right? Emperically, how does the proposed model perform on such datasets?**
>
> A4: Actually, for datasets with little periodicity, LightGTS is insensitive to the prediction of trends with the selection of patch size, and the the model size would be affected by the selection of patch size. Take the exchange dataset which exhibits little periodicity as an example, the performance of LightGTS in zero-shot setting remains stable when the patch size is set to 1, 4, 6, 8, 16 and 24, respectively. In the paper, we use fft-extracted periods: 6.
>
> | Patch size | 1(**point patch**) | 4 | 6 (fft-extracted) | 8 | 16 | 24 |
> | --- | --- | --- | --- | --- | --- | --- |
> | Exchange-MSE | 0.347 | 0.348 | 0.347 | 0.353 | 0.351 | 0.354 |
>
> **Q5: How to deal with different cycle lengths.**
>
> A5: For the case where a time series is known to have different cycle lengths, we pick the smallest cycle as the patch size. For example, traffic and electricity dataset both have two cycle lengths, 24 (one day) and 168 (one week), and we choose 24 as the patch size. For the case where the cycle length is not well known, FFT can be used for cycle length extraction, which we have discussed in the Appendix B.1.

---

> > ### Comment · Reviewer_AuKx · 2025-04-03
> >
> > Thank you for the rebuttal. Many of my concerns have been addressed. But my major concerns remain, the core component of the proposed LightGTS, are only designed for periodical time series data, why it should be working for dataset without periodicity? The authors should explain this observation. The authors provide results only on one dataset without periodicity (Exchange), with predicating length from 96 to 720 (more than 2 years), the authors should evaluate on more datasets without periodicity in a more realistic scenario.

---

> > > ### Author Response · Authors · 2025-04-04
> > >
> > > Thank you for your thoughtful questions and valuable feedback. We appreciate the opportunity to clarify and refine our responses.
> > >
> > > - Our design of periodical patching enables the model to naturally adapt to time series with different granularity and periodicity. However, this does not imply that we focus only on periodic information. By computing the attention between patch tokens, the model also learns the overall trend of time series, which is key to making the LightGTS work for datasets without periodicity. Moreover, our non-autoregressive decoder avoids cumulative errors, thus enabling more accurate prediction of trends.
> > > - Experiments: Actually, the Chronos Benchmark II presented in Q1 includes 27 datasets, and 10 of them are datasets without periodicity. Based on your suggestion, we supplement the evaluation of the following three datasets without periodicity on four prediction lengths: 24, 36, 48, and 60 in zero-shot and full-shot setting. Our experimental setup follows TFB [1], and since these datasets are of daily granularity, predicting values up to two months is consistent with practical application scenarios.
> > >
> > >     zero-shot setting:
> > >
> > >     | Models | LightGTS | TimesFM | Timer | MOIRAI |
> > >     | --- | --- | --- | --- | --- |
> > >     | Metrics | MSE / MAE | MSE / MAE | MSE / MAE | MSE / MAE |
> > >     | nn5 | **0.768/ 0.599** | 0.780/ 0.605 | 1,260/ 0.878 | 0.786/ 0.623 |
> > >     | wike2000 | 597.043/ 1.358 | **475.582/ 1.077** | 605.070 / 1.428 | 512.875 / 1.209|
> > >     | NASDAQ | **0.885/ 0.681** | 1.034/ 0.687  | 0.890/ 0.688 | 1.045/ 0.701 |
> > >
> > >     full-shot setting:
> > >
> > >     | Models | LightGTS | PatchTST | iTransformer | FITS | TimeMixer | Pathformer | PDF |
> > >     | --- | --- | --- | --- | --- | --- | --- | --- |
> > >     | Metrics | MSE / MAE | MSE / MAE | MSE / MAE | MSE / MAE | MSE / MAE | MSE / MAE | MSE / MAE |
> > >     | nn5 | **0.648/ 0.543** | 0.692/ 0.594 | 0.660/ 0.550 | 0.811/ 0.653 | 0.656/ 0.556 | 0.698/ 0.582 | 0.657/ 0.554 |
> > >     | wike2000 | **511.017/ 1.128** | 513.966/ 1.140 | 545.647/ 1.189 | 732.541/ 1.486 | 518.679/ 1.251 | 788.620/ 1.527 | 543.558/ 1.170 |
> > >     | NASDAQ | **0.882/ 0.677** | 0.972/ 0.721 | 0.944/ 0.683 | 1.043/ 0.774 | 1.005/ 0.731 | 1.014/ 0.726 | 1.021/ 0.724 |
> > >
> > > [1] Qiu, Xiangfei, et al. "Tfb: Towards comprehensive and fair benchmarking of time series forecasting methods." arXiv preprint arXiv:2403.20150 (2024).

---

### Official Review · Reviewer_3dom · 2025-03-13

**Overall Recommendation:** 4

**Summary:**

The paper proposes a general purpose pretrained time series forecasting model called LightGTS. The model is an encoder-decoder transformer operating on patches of time series observations. However, unlike existing works which operate on fixed patch lengths, LightGTS uses a dynamically adjusted patch length which is aligned to the seasonality of the time series. Authors argue that this enables better modeling of time series data compared to an arbitrarily selected fixed patch length. Empirical results have been reported on 9 time series datasets.

**Claims And Evidence:**

The central claim of the paper is that LightGTS is a general purpose pretrained time series model which is more accurate and lightweight compared to existing pretrained models. Authors argue that "existing TSFMs largely depends on massive pre-training data and large model parameters".

**Existing models depend on large scale training and large number of parameters**: Although this is true for most existing models, in the end, LightGTS is also pretrained on a large scale multi-source corpus. However, the claim about fewer number of parameters is correct.

**LightGTS is more accurate compared to existing TSFMs**: This claim is not supported by the evaluation conducted in the paper. In particular, the evaluation is severely limited to draw meaningful conclusions. The experiment design also suffers from serious flaws that have previously been discussed in the community. Note that these flaws are not unique to LightGTS and have also existed in previous works. However, with the rapid progress in the field and existence of better benchmarks, it is imperative that newer works do a better job at evaluation, especially when the claim involves a general purpose pretrained model. Please see **Methods And Evaluation Criteria** for details.

**Fixed patch embedding does not transfer well to new seasonalities**: The experiment in Fig 2 demonstrates this to a certain degree and I intuitively agree with this assertion. However, more evidence is needed showing that this is an issue in existing patch-based pretrained models such as Chronos-Bolt and TimesFM. I also have questions about Fig 2. Please refer to **Questions for Authors**.

**Essential References Not Discussed:**

Some works such as TinyTimeMixers have made similar claims as LightGTS and should be discussed as part of related work.

**Experimental Designs Or Analyses:**

Please see **Methods and Evaluation Criteria**.

**Methods And Evaluation Criteria:**

The authors propose a reasonable architecture for a general purpose time series model. I particularly find the contribution of periodical patching quite interesting. I would encourage the authors to set the stage better for their approach, e.g., by discussing Moirai better. Moirai uses multiple patch sizes and a patch selection mechanism with a similar motivation to LightGTS. Periodical patching is a natural and elegant way to address the limitation of fixed patch sizes.

The breadth of experiments is severely limited for a general purpose model. Only 9 datasets have been studied, which is not enough to justify the title of a general purpose model. Moreover, 5 of these datasets belong to the same domain with 4 being essentially the same dataset (ETTh1, ETTh2, ETTm1, ETTm2). This infamous "ETT long term forecasting benchmark" is often criticized for its flaws such as limited domain coverage and the practice of forecasting at unreasonable horizons (e.g., 720 days into the future for exchange rate or oil temperature of a transformer at a specific hour months into the future). Every new model somehow beats this benchmark; however, there is still barely any absolute progress, only an illusion of it. Please refer to the talk (and paper) from Christoph Bergmeir [1, 2] where he discusses the limitation of this benchmark and current evaluation practices. A _very recent_ position paper [3] also conducted a comprehensive evaluation of models on this benchmark showing that there's no obvious winner.

One (not so difficult) way to improve the quality of evaluation is to include results on better benchmarks that have been proposed recently in the context of pretrained time series models.

- Chronos Benchmark II: This benchmark includes 27 datasets (42, if you include Benchmark I) providing a comprehensive coverage over domains, frequencies and other properties. Please refer to https://github.com/autogluon/fev for details on how to use the benchmark.
- GIFT-Eval: This benchmark includes 90+ tasks across multiple datasets and domains. It also provides a leaderboard of existing pretrained models. Please refer to https://github.com/SalesforceAIResearch/gift-eval.

[1] https://neurips.cc/virtual/2024/workshop/84712#collapse108471
[2] Hewamalage, Hansika, Klaus Ackermann, and Christoph Bergmeir. "Forecast evaluation for data scientists: common pitfalls and best practices." Data Mining and Knowledge Discovery 37.2 (2023): 788-832.
[2] Brigato, Lorenzo, et al. "Position: There are no Champions in Long-Term Time Series Forecasting." arXiv preprint arXiv:2502.14045 (2025).

**Other Comments Or Suggestions:**

Authors use new terms for known quantities in time series forecasting. For example, _scale_ is used to refer to _frequency_ and _cycle length_ is used to refer to _seasonality_. I would encourage the authors to use existing terms to better align with the literature.

**Other Strengths And Weaknesses:**

LightGTS is only a point forecaster, whereas most existing pretrained models (Chronos, Moirai, TimesFM, Chronos-Bolt) support probabilistic forecasting. Uncertainty quantification is critical feature for downstream decision making based on a forecast.

**Questions For Authors:**

- For the experiment in Fig 2, did the authors only trained the model on ETTh1 dataset? If yes, I find it very surprising that it transfers to very different types of seasonal patterns such as those in the solar dataset. Is the claim here that only pretraining on a single seasonality could somehow deliver a pretrained model? Out of curiosity, how does the model only trained with a single seasonality perform quantitatively? Why does the pretraining corpus contain diverse seasonalities and how were these selected?

**Relation To Broader Scientific Literature:**

The paper makes a contribution to the rapidly evolving area of pretrained time series forecasting models. The idea of periodical patching is promising and addresses the issue of fixed patch sizes in previous models. The evaluation is limited and can be improved by the inclusion of better benchmarks.

**Theoretical Claims:**

The paper does not make theoretical contributions. Also, **Theorem 3.1** is not a theorem per se but more of a remark.

---

> ### Author Rebuttal · Authors · 2025-04-01
>
> **Q1: Improve the quality of evaluation**
>
> A1: We use Chronos Benchmark II you mentioned to further evaluate the generalizability of LightGTS. As shown in the table below, LightGTS shows outstanding performance, second only to Moirai-large whose training corpus overlap is much higher. In addition, LightGTS also exhibits superior efficiency compared to other foundation model.
>
> | Model | Moirai-large | Chronos-large | TimesFM | LightGTS_mini(ours) | Seasonal Naive |
> | --- | --- | --- | --- | --- | --- |
> | Average Relative Error | **0.791** | 0.824 | 0.882 | 0.819 | 1.000 |
> | Median inference time (s) | 14.254 | 26.827 | 0.396 | 1.390 | **0.096** |
> | Training corpus overlap (%) | 81.5% | 0% | 14.8% | 31% | 0% |
>
> **Q2: Periodical Patching vs. Multi-Patching**
>
> A2:  While MOIRAI's predefined patch sizes based on sampling frequency offer some solution for  consistent modeling across different frequencies, they are still fixed and lack flexibility in certain scenarios. In contrast, Periodical Patching adaptively divides patches according to scale-invariant periodicity, enabling more flexible and unified modeling for datasets with varying frequencies.
>
> **Q3: Some works such as TinyTimeMixers have made similar claims as LightGTS and should be discussed as part of related work.**
>
> A3: Thank you for mentioning TTMs, which are also lightweight TSFMs like LightGTS. We will differentiate LightGTS from TTMs in the following two aspects:
>
> - **Flexibility**: TTMs have fixed input and output formats, which imposes limitations in downstream applications. In contrast, LightGTS supports flexible input and output configurations.
> - **Adaptive Patching**: While TTMs employ adaptive patching through CV-inspired patch merging techniques to capture multi-scale features, they remain constrained by predefined patch sizes. LightGTS, however, leverages periodical patching that adaptively segments time series based on the intrinsic periods. This approach enables LightGTS to achieve unified modeling across datasets with varying scales.
>
> We would update it in the related work of the revised paper.
>
> **Q4: LightGTS is only a point forecaster, whereas most existing pretrained models (Chronos, Moirai, TimesFM, Chronos-Bolt) support probabilistic forecasting. Uncertainty quantification is critical feature for downstream decision making based on a forecast.**
>
> A4: We are developing an enhanced version of LightGTS with quantile regression pre-train task to support probabilistic forecasting (**uncertainty quantification**), which will be released in future updates.
>
> **Q5: Replace non-standard terms (e.g., "scale" for frequency, "cycle length" for seasonality) with established conventions.**
>
> A5:  We sincerely appreciate your feedback regarding terminology alignment. In the revised manuscript, we will systematically replace non-standard terms with established terminology from time series.
>
> **Q6: Was the model only trained on the ETTh1 dataset? How does the model perform quantitatively when trained on a single seasonality?Why does the pretraining corpus include diverse seasonalities, and how were they selected?**
>
> A6:
>
> - Yes, we found that pre-training on a single seasonality dataset can enable effective transfer learning.
> - **Experiment:** LightGTS-single was pre-trained on ETTh1 and directly tested on downstream datasets in a zero-shot setting. As shown in the table below, LightGTS-single achieved strong transfer performance on datasets with single seasonality (e.g., Solar), while its performance was less optimal on datasets with multiple seasonalities (e.g., Electricity). By pre-training on multi-seasonalities datasets, LightGTS-mini significantly improved its performance on datasets with multiple seasonalities.
>
> | Datasets | ETTh2 | ETTm2 | Weather | Solar | Electricity |
> | --- | --- | --- | --- | --- | --- |
> | Metrics | MSE/MAE | MSE/MAE | MSE/MAE | MSE/MAE | MSE/MAE |
> | PatchTST | 0.351/0.395 | 0.256/0.314 | 0.224/0.261 | 0.200/0.284 | 0.171/0.270 |
> | LightGTS-single | 0.357/0.381 | 0.251/0.314 | 0.232/0.274 | 0.227/302 | 0.246/0.358 |
> | LightGTS-mini | 0.359/0.396 | 0.250/0.318 | 0.210/0.258 | 0.196/0.269 | 0.214/0.306 |
> - **Pre-training corpus selection:** We did not explicitly filter datasets based on seasonality. Instead, we collected datasets from diverse domains (e.g., energy, weather, traffic), assuming that time-series data from different domains inherently contain distinct seasonal patterns. This approach ensures broader generalizability while retaining domain-specific characteristics.

---

> > ### Comment · Reviewer_3dom · 2025-04-02
> >
> > Thank you!
> >
> > This is one of the best rebuttals I have read today: straight to the point. I truly appreciate your efforts on expanding the evaluation. As such, I am more confident updating my score to 4. Thank you also for clarifying on the other points that I had raised.
> >
> > A small note, but this is more of request: In the final version of the paper, could you also add evaluations on other benchmarks such as GIFT-Eval? This will have a positive cascading effect on the community and hopefully everyone will start doing better evaluations. It may also be a good idea to include more recent models such as Chronos-Bolt, TabPFN-TS and TimesFM-2.0. Note that these models fall within the concurrent work guidelines for ICML, but it may be a good idea to include them for completeness.

---

> > > ### Author Response · Authors · 2025-04-04
> > >
> > > Dear Reviewer 3dom,
> > >
> > > Thank you sincerely for your thoughtful feedback and constructive suggestions. We're deeply encouraged by your recognition of our rebuttal efforts and are committed to implementing your recommendations in the final version. Specifically:
> > >
> > > - **Expanded Evaluations**: We will include additional benchmarks like GIFT-Eval to strengthen the empirical analysis, aiming to promote more comprehensive evaluations in the time-series community.
> > > - **Model Comparisons**: We will incorporate evaluations of recent models such as Chronos-Bolt, TabPFN-TS, and TimesFM-2.0 where feasible.
> > >
> > > Your insights have been invaluable in refining this work, and we fully agree that rigorous benchmarking benefits the broader research ecosystem.
> > >
> > > Best regards,
> > >
> > > Authors

---

### Decision · Program_Chairs · 2025-05-01

**Decision:**

Accept (poster)

**Comment:**

Contributions
[1] The paper presents LightGTS, a pre-training time-series forecasting model that works on variable patch lengths. The patch lengths are aligned to the seasonality of time-series. [Reviewer 3dom]
[2] Proposed a new tokenization method for time series datasets utilizing periodical information [Reviewer AuKx]
[3] It proposes a periodical tokenization, which adaptively splits time series into patches aligned with intrinsic periods to handle varying scales, and periodical parallel decoding [Reviewer CW8j]
[4] The significant contribution of this paper is very similar to that of the article, which also puts forward an Adaptive patching technology. such as Tiny Time Mixers (TTMs) [Reviewer Kj5L]

Weakness:
[1] The paper claims a foundation model with fewer parameters, and demonstrates that the fixed patch embedding does not transfer well to new seasonalities [Reviewer 3dom]
[2] To the comment of [Reviewer AuKx] that the paper only addresses periodic time-series data, authors have demonstrated the method on some non-periodic data sets during rebuttal and claim that “By computing the attention between patch tokens, the model also learns the overall trend of time series”. An ablation study to support the claim that  parallel decoding helps with trend forecasting is important.
[3] In response to [Reviewer Kj5L] comments on comparison with TTMs, author have made a comparative study with TTMs-advanced.